# Modelling the neural code in large populations of correlated neurons

**Sacha Sokoloski[1,2]\*, Amir Aschner[3], Ruben Coen-Cagli[1,3]**

[1]Department of Systems and Computational Biology, Albert Einstein College of Medicine, Bronx, United States; [2]Institute for Ophthalmic Research, University of Tübingen, Tübingen, Germany; [3]Dominick P. Purpura Department of Neuroscience, Albert Einstein College of Medicine, Bronx, United States

**Abstract** Neurons respond selectively to stimuli, and thereby define a code that associates stimuli with population response patterns. Certain correlations within population responses (noise correlations) significantly impact the information content of the code, especially in large populations. Understanding the neural code thus necessitates response models that quantify the coding properties of modelled populations, while fitting large-scale neural recordings and capturing noise correlations. In this paper, we propose a class of response model based on mixture models and exponential families. We show how to fit our models with expectation-maximization, and that they capture diverse variability and covariability in recordings of macaque primary visual cortex. We also show how they facilitate accurate Bayesian decoding, provide a closed-form expression for the Fisher information, and are compatible with theories of probabilistic population coding. Our framework could allow researchers to quantitatively validate the predictions of neural coding theories against both large-scale neural recordings and cognitive performance.

## Introduction

A foundational idea in sensory neuroscience is that the activity of neural populations constitutes a 'neural code' for representing stimuli (*Dayan and Abbott, 2005*; *Doya, 2007*): the activity pattern of a population in response to a sensory stimulus encodes information about that stimulus, and downstream neurons decode, process, and re-encode this information in their own responses. Sequences of such neural populations implement the elementary functions that drive perception, cognition, and behaviour (*Pitkow and Angelaki, 2017*). Therefore, by studying the encoding and decoding of population responses, researchers may investigate how information is processed along neural circuits, and how this processing influences perception and behaviour (*Wei and Stocker, 2015*; *Panzeri et al., 2017*; *Kriegeskorte and Douglas, 2018*).

Given a true statistical model of how a neural population responds to (encodes information about) stimuli, Bayes' rule can transform the encoding model into an optimal decoder of stimulus information (*Zemel et al., 1998*; *Pillow et al., 2011*). However, when validated as Bayesian decoders, statistical models of neural encoding are often outperformed by models trained to decode stimulus-information directly, indicating that the encoding models miss key statistics of the neural code (*Graf et al., 2011*; *Walker et al., 2020*). In particular, the correlations between neurons' responses to repeated presentations of a given stimulus (noise correlations), and how these noise correlations are modulated by stimuli, can strongly impact coding in neural circuits (*Zohary et al., 1994*; *Abbott and Dayan, 1999*; *Sompolinsky et al., 2001*; *Ecker et al., 2016*; *Kohn et al., 2016*; *Schneidman, 2016*), especially in large populations of neurons (*Moreno-Bote et al., 2014*; *Montijn et al., 2019*; *Bartolo et al., 2020*; *Kafashan et al., 2021*; *Rumyantsev et al., 2020*).

Statistically validating theories of population coding in large neural circuits thus depends on encoding models that support accurate Bayesian decoding, effectively capture noise-correlations,

\*For correspondence:
sacha.sokoloski@mailbox.org

**Competing interests:** The authors declare that no competing interests exist.

and efficiently fit large-scale neural recordings. There are at least two classes of neural recordings for which established models have facilitated such analyses. Firstly, for recordings of binary spike-counts, pairwise-maximum entropy models (*Schneidman et al., 2006*; *Lyamzin et al., 2010*; *Granot-Atedgi et al., 2013*; *Tkačik et al., 2013*; *Meshulam et al., 2017*; *Maoz et al., 2020*) have been used to investigate the structure of the retinal code (*Ganmor et al., 2015*; *Tkačik et al., 2015*). Secondly, when modelling dynamic spike-train recordings, generalized linear models (GLMs) have proven effective at modelling spatio-temporal features of information processing in the retina and cortex (*Pillow et al., 2008*; *Park et al., 2014*; *Runyan et al., 2017*; *Ruda et al., 2020*).

Nevertheless, many theories of neural coding are formulated in terms of unbounded spike-counts (*Ma et al., 2006*; *Beck et al., 2011a*; *Ganguli and Simoncelli, 2014*; *Makin et al., 2015*; *Yerxa et al., 2020*), rather than the binary spike-counts of pairwise maximum entropy models. Furthermore, neural correlations are often low-dimensional (*Arieli et al., 1996*; *Ecker et al., 2014*; *Goris et al., 2014*; *Rabinowitz et al., 2015*; *Okun et al., 2015*; *Semedo et al., 2019*), in contrast with the correlations that result from the fully connected, recurrent structure of standard GLMs. Although there are extensions of the GLM approach that capture shared-variability (*Vidne et al., 2012*; *Archer et al., 2014*; *Zhao and Park, 2017*), they seem unable to support exact Bayesian decoding. Similarly, methods such as factor analysis that model unbounded spike-counts as continuous variables have proven highly effective at modelling neural correlations in large-scale recordings (*Yu et al., 2009*; *Cunningham and Yu, 2014*; *Ecker et al., 2014*; *Semedo et al., 2019*), yet it is also unknown if they can support accurate Bayesian decoding.

Towards modelling spike-count responses and accurate Bayesian decoding in large populations of correlated neurons, we develop a class of encoding model based on finite mixtures of Poisson distributions. Within neuroscience, Poisson mixtures are widely applied to modelling the spike-count distributions of individual neurons (*Wiener and Richmond, 2003*; *Shidara et al., 2005*; *Goris et al., 2014*; *Taouali et al., 2016*). Outside of neuroscience, mixtures of multivariate Poisson distributions are an established model of multivariate count distributions that effectively capture correlations in count data (*Karlis and Meligkotsidou, 2007*; *Inouye et al., 2017*).

Building on the theory of exponential family distributions (*Wainwright and Jordan, 2008*; *Macke et al., 2011b*), our model extends previous mixture models of multivariate count data in two ways. Firstly, we develop a tractable extension of Poisson mixtures that captures both over- and under-dispersed response variability (i.e. where the response variance is larger or smaller than the mean, respectively) based on Conway-Maxwell Poisson distributions (*Shmueli et al., 2005*; *Stevenson, 2016*). Secondly, we introduce an explicit dependence of the model on a stimulus variable, which allows the model to accurately capture changes in response statistics (including noise correlations) across stimuli. Importantly, the resulting encoding model affords closed-form expressions for both its Fisher information and probability density function, and thereby a rigorous quantification of the coding properties of a modelled neural population (*Dayan and Abbott, 2005*). Moreover, the model learns low-dimensional representations of stimulus-driven neural activity, and we show how it captures a fundamental property of population codes known as information-limiting correlations (*Moreno-Bote et al., 2014*; *Montijn et al., 2019*; *Bartolo et al., 2020*; *Kafashan et al., 2021*; *Rumyantsev et al., 2020*).

We apply our mixture model framework to both synthetic data and recordings from macaque primary visual cortex (V1), and demonstrate that it effectively models responses of populations of hundreds of neurons, captures noise correlations, and supports accurate Bayesian decoding. Moreover, we show how our model is compatible with the theory of probabilistic population coding (*Zemel et al., 1998*; *Pouget et al., 2013*), and could thus be used to study the theoretical coding properties of neural circuits, such as their efficiency (*Ganguli and Simoncelli, 2014*), linearity (*Ma et al., 2006*), or information content (*Moreno-Bote et al., 2014*).

## Results

A critical part of our theoretical approach is based on expressing models of interest in exponential family form. An exponential family distribution $p(n)$ over some data $n$ (in our case, neural responses) is defined by the proportionality relation $p(n) \propto e^{\boldsymbol{\theta} \cdot \mathbf{s}(n)} b(n)$, where $\boldsymbol{\theta}$ are the so-called natural parameters, $\mathbf{s}(n)$ is a vector-valued function of the data called the sufficient statistic, and $b(n)$ is a scalar-valued, non-negative function called the base measure (*Wainwright and Jordan, 2008*). The

exponential family form allows us to modify and extend existing models in a simple and flexible manner, and derive analytical results about the coding properties of our models. We demonstrate our approach with applications to both synthetic data, and experimental data recorded in V1 of anaesthetized and awake macaques viewing drifting grating stimuli at different orientations (for details see Materials and methods).

## Extended Poisson mixture models capture spike-count variability and covariability

Our first goal is to define a class of models of stimulus-independent, neural population activity, that model neural activity directly as spike-counts, and that accurately capture single-neuron variability and pairwise covariability. We base our models on Poisson distributions, as they are widely applied to modelling the trial-to-trial distribution of the number of spikes generated by a neuron (*Dayan and Abbott, 2005*; *Macke et al., 2011a*). We will also generalize our Poisson-based models with the theory of Conway-Maxwell (CoM) Poisson distributions (*Sur et al., 2015*; *Stevenson, 2016*; *Chanialidis et al., 2018*). The two-parameter CoM-Poisson model contains the one-parameter Poisson model as a special case, however, whereas the Poisson model always has a Fano factor (FF; the variance divided by the mean) of 1, the CoM-Poisson model can exhibit both over- (FF>1) and under-dispersion (FF<1), and thus capture the broader range of Fano factors observed in cortex (*Stevenson, 2016*).

The other key ingredient in our modelling approach are mixtures of Poisson distributions, which have been used to model complex spike-count distributions in cortex, and also allow for over-dispersion (*Shidara et al., 2005*; *Goris et al., 2014*; *Taouali et al., 2016*; *Figure 1A*). In our case, we mix multiple, independent Poisson distributions in parallel, as such models can capture covariability in count data as well (see *Karlis and Meligkotsidou, 2007* for a more general formulation of multivariate Poisson mixtures than what we consider here). To construct such a model, we begin with a product of independent Poisson distributions (IP distribution), one per neuron. We then mix a finite number of component IP models, to arrive at a multivariate spike-count, finite mixture model (see Materials and methods). Importantly, although each component of this mixture is an IP distribution, randomly switching between components induces correlations between the neurons (*Figure 1B,C*).

IP mixtures can in fact model arbitrary covariability between neurons (see Materials and methods, *Equation 7*); however, they are still limited because the model neurons in an IP mixture are always over-dispersed. To overcome this, it is helpful to consider factor analysis (FA), which is widely applied to modelling neural population responses (*Cunningham and Yu, 2014*). IP mixtures are similar to FA, in that FA represents the covariance matrix of neural responses as the sum of a diagonal matrix that helps capture individual variance, and a low-rank matrix that captures covariance (see *Bishop, 2006*), and FA and IP mixtures can be fine-tuned to capture covariance arbitrarily well. However, whereas FA has distinct parameters for representing means and diagonal variances, the means and variances in an IP mixture are coupled through shared parameters (see Materials and methods, *Equation 6*). Our strategy will thus be to break this coupling between means and variances by granting IP mixtures an additional set of parameters based on the theory of CoM-Poisson distributions.

To do so, we first show how to express an IP mixture as the marginal distribution of an exponential family distribution. Note that an IP mixture with $d_K$ components may be expressed as a latent variable model over spike-count vectors $\boldsymbol{n}$ and latent component-indices $k$, where $1 \leq k \leq d_K$. In this formulation we denote the $k$th component distribution by $p(\boldsymbol{n} \mid k)$, and the probability of realizing (switching to) the $k$th component by $p(k)$. The mixture model over spike-counts $\boldsymbol{n}$ is then expressed as the marginal distribution $p(\boldsymbol{n}) = \sum_{k=1}^{d_K} p(\boldsymbol{n} \mid k)p(k) = \sum_{k=1}^{d_K} p(\boldsymbol{n}, k)$, of the joint distribution $p(\boldsymbol{n}, k)$. Under mild regularity assumptions (see Materials and methods), we may reparameterize this joint distribution in exponential family form as

$$p(\boldsymbol{n}, k) \propto \frac{e^{\boldsymbol{\theta}_N \cdot \boldsymbol{n} + \boldsymbol{\theta}_K \cdot \boldsymbol{\delta}(k) + \boldsymbol{n} \cdot \boldsymbol{\Theta}_{NK} \cdot \boldsymbol{\delta}(k)}}{\prod_{i=1}^{d_N} n_i!}, \tag{1}$$

where the vectors $\boldsymbol{\theta}_N$ and $\boldsymbol{\theta}_K$, and matrix $\boldsymbol{\Theta}_{NK}$ are the natural parameters of $p(\boldsymbol{n}, k)$, and $\boldsymbol{\delta}(k) = (\delta_2(k), \ldots, \delta_{d_K}(k))$ is the Kronecker delta vector defined by $\delta_j(k) = 1$ if $j = k$, and 0 otherwise.

This representation affords an intuitive interpretation. In general, the natural parameters of an IP distribution are the logarithms of the average spike-counts (firing rates), and the natural parameters

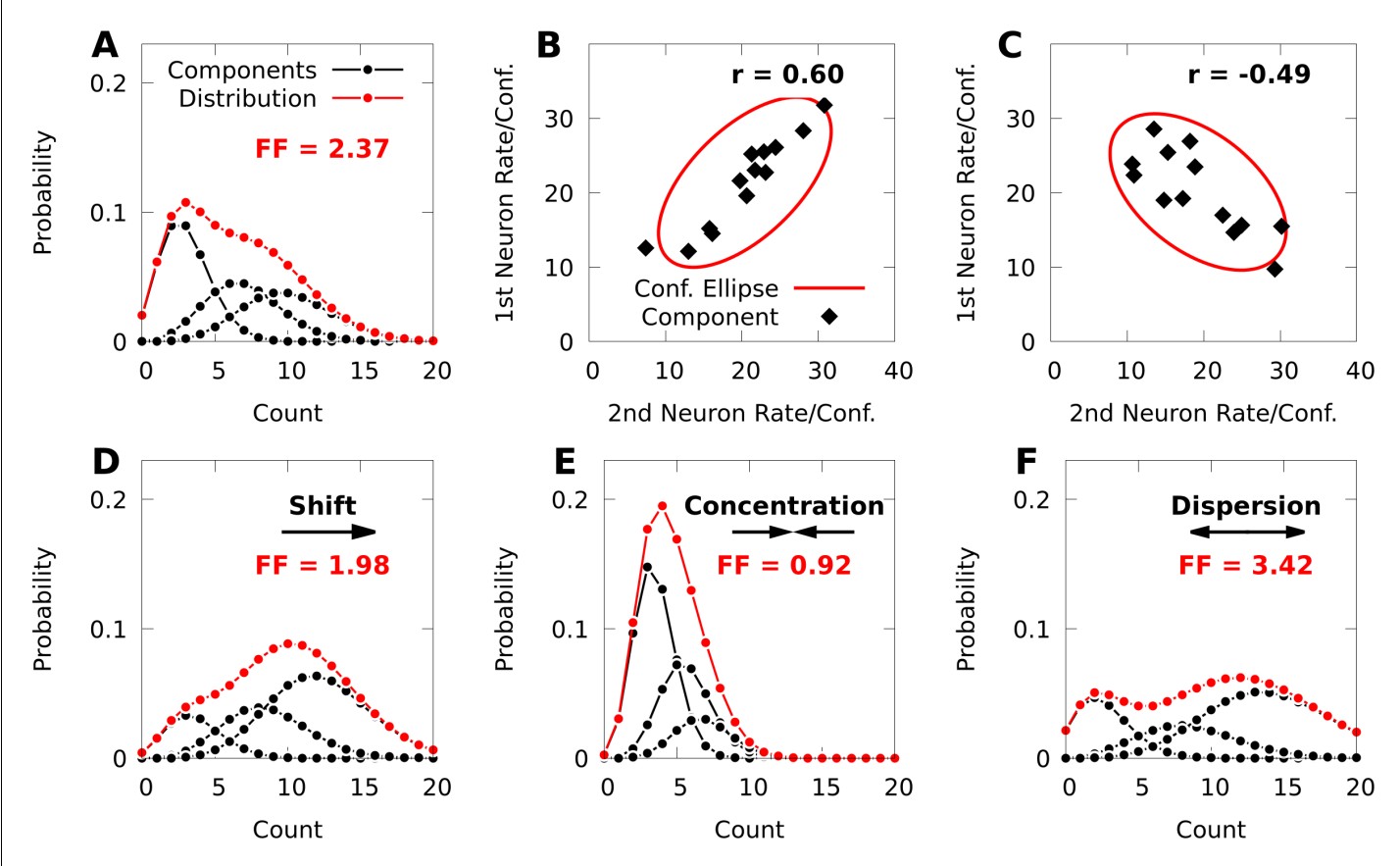

**Figure 1.** Poisson mixtures and Conway-Maxwell extensions exhibit spike-count correlations, and over- and under-disperson. (A) A Poisson mixture distribution (red), defined as the weighted sum of three component Poisson distributions (black; scaled by their weights). FF denotes the Fano Factor (variance over mean) of the mixture. (B, C) The average spike-count (rate) of the first and second neurons for each of 13 components (black dots) of a bivariate IP mixture, and 68% confidence ellipses for the spike-count covariance of the mixture (red lines; see *Equations 6 and 7*). The spike-count correlation of each mixture is denoted by *r*. (D) Same model as A, except we shift the distribution by increasing the baseline rate of the components. (E, F) Same model as A, except we use an additional baseline parameter based on Conway-Maxwell Poisson distributions to concentrate (E) or disperse (F) the mixture distribution and its components.

of the first component distribution $p(\boldsymbol{n} \mid k = 1)$ of an IP mixture are simply $\boldsymbol{\theta}_N$. The natural parameters of the $k$ th component for $k>1$ are then the sum of the 'baseline' parameters $\boldsymbol{\theta}_N$ and column $k-1$ from the matrix of parameters $\boldsymbol{\Theta}_{NK}$ (*Equation 13*, Materials and methods). Because the dimension of the baseline parameters $\boldsymbol{\theta}_N$ is much smaller than the total number of parameters in a given mixture, the baseline parameters provide a relatively low-dimensional means of affecting all the component distributions of the given mixture, as well as the probability distribution over indices $p(k)$ (*Figure 1D*; see Materials and methods, *Equation 12* for how the index-probabilities $p(k)$ depend on $\boldsymbol{\theta}_N$).

We next extend *Equation 1* with the theory of CoM-Poisson distributions, and define the latent variable exponential family

$$p(\mathbf{n},k) \propto e^{\boldsymbol{\theta}_N \cdot \mathbf{n} + \boldsymbol{\theta}_N^* \cdot \mathbf{lf}(\mathbf{n}) + \boldsymbol{\theta}_K \cdot \boldsymbol{\delta}(k) + \mathbf{n} \cdot \boldsymbol{\Theta}_{NK} \cdot \boldsymbol{\delta}(k)}, \tag{2}$$

where $\mathbf{lf}(\mathbf{n}) = (\log(n_1!), \dots, \log(n_{d_N}!))$ is the vector of log-factorials of the individual spike-counts, and $\boldsymbol{\theta}_N^*$ are a set of natural parameters derived from CoM-Poisson distributions (see Materials and methods). Based on this construction, each component $p(\mathbf{n} \mid k)$ is a product of independent CoM-Poisson distributions, and when $\boldsymbol{\theta}_N^* = -\mathbf{1}$, we recover an IP mixture defined by *Equation 1* with parameters $\boldsymbol{\theta}_N$, $\boldsymbol{\theta}_K$, and $\boldsymbol{\Theta}_{NK}$. The first component of this model $p(\mathbf{n} \mid k = 1)$ has parameters $\boldsymbol{\theta}_N$ and $\boldsymbol{\theta}_N^*$, and as with the IP mixture, the parameters $\boldsymbol{\theta}_N$ are translated by column $k-1$ of $\boldsymbol{\Theta}_{NK}$ when $k>1$. However, the parameters $\boldsymbol{\theta}_N^*$ are never translated, and remain the same for each component distribution

(*Equation 16*, Materials and methods, and see *Equation 15* for formulae for the index-probabilities $p(k)$). We refer to models defined by *Equation 2* as CoM-based (CB) mixtures, and $\boldsymbol{\theta}_N^*$ as CB parameters.

Due to the addition of the CB parameters $\boldsymbol{\theta}_N^*$, a CB mixture breaks the coupling between the spike-count means and variances that is present in the simpler IP mixture (*Equation 17*, Materials and methods). In *Figure 1D–F*, we demonstrate how changing the parameters of a CB mixture can concentrate or disperse both the mixture distribution and its components, and that a CB mixture can indeed exhibit under-dispersion.

To validate our mixture models, we tested if they capture variability and covariability of V1 population responses to repeated presentations of a grating stimulus with fixed orientation ($d_N = 43$ neurons and $d_T = 355$ repetitions of 150 ms duration in one awake macaque; $d_N = 70$ and $d_T = 1,200$ of duration 70 ms in one anaesthetized macaque). We fit our mixtures to the complete datasets with expectation-maximization (EM, a standard choice for training finite mixture models [*McLachlan et al., 2019*] see Materials and methods). The CB mixture accurately captured single-neuron variability (*Figure 2A–B*, red symbols), including both cases of over-dispersion and under-dispersion. On the other hand, the simpler IP mixture (*Figure 2A–B*, blue symbols) cannot accommodate under-dispersion due to its mathematical limits, and demonstrated limited ability to model over-dispersion due to the coupling between the mean and variance (*Equation 6*).

To understand how the CB parameters allow the CB mixture to overcome the limits of the IP mixture, we plot a histogram of the CB parameters $\boldsymbol{\theta}_N^*$ for both fits (*Figure 2C–D*). If the CB parameter of a given CoM-Poisson distribution is $< -1$, $> -1$, or $= -1$, then the CoM-Poisson distribution is under-dispersed, over-dispersed, or Poisson-distributed, respectively. When a CB mixture is fit to the awake data (*Figure 2C*), we see that it learns a range of values for the CB parameters around $-1$, to accommodate the variety of Fano factors observed in the awake data (*Figure 2A*). On the anaesthetized data, even though IP mixtures can capture over-dispersion, the IP mixture underestimates the dispersion of neurons due to the coupling between the mean and variance (*Figure 2B*). The CB mixture thus uses the CB parameters to further disperse its model neurons (*Figure 2D*).

In contrast with individual variability, we found that both mixture models were flexible enough to qualitatively capture pairwise noise correlation structure in both awake and anaesthetized animals (*Figure 3A–B*), and that the distributions of modelled neural correlations were broadly similar when compared to the data (*Figure 3C–D*). In Appendix 1, we rigorously compare IP mixtures, CB mixtures, and FA on our datasets, and show that although FA is better than our mixture models at capturing second-order statistics in training data, IP mixtures and CB mixtures achieve comparable predictive performance as FA when evaluated on held-out data.

## Extended Poisson mixture models capture stimulus-dependent response statistics

So far, we have introduced the exponential family theory of IP and CB mixtures, and shown how they capture response variability and covariability for a fixed stimulus. To allow us to study stimulus encoding and decoding, we further extend our mixtures by inducing a dependency of the model parameters on a stimulus. When there are a finite number of stimulus conditions and sufficient data, we may define a stimulus-dependent model with a lookup table, and fit it by fitting a distinct model at each stimulus condition. However, this is inefficient when the amount of data at each stimulus-condition is limited and the stimulus-dependent statistics have structure that is shared across conditions. A notable feature of the exponential family parameterizations in *Equations 1 and 2* is that the baseline parameters influence both the index probabilities and all the component distributions of the model. This suggests that by restricting stimulus-dependence to the baseline parameters, we might model rich stimulus-dependent response structure, while bounding the complexity of the model.

In general, we refer to any finite mixture with stimulus-dependent parameters as a conditional mixture (CM), and depending on whether the CM is based on *Equations 1 and 2* and, we refer to it as an IP- or CB-CM, respectively. Although there are many ways we might induce stimulus-dependence, in this paper we consider two forms of CM: (i) a maximal CM, which we implement as a lookup table, such that all the parameters in *Equations 1 and 2* and depend on the stimulus, and (ii) a minimal CM, for which we restrict stimulus-dependence to the baseline parameters $\boldsymbol{\theta}_N$. This results in the CB-CM

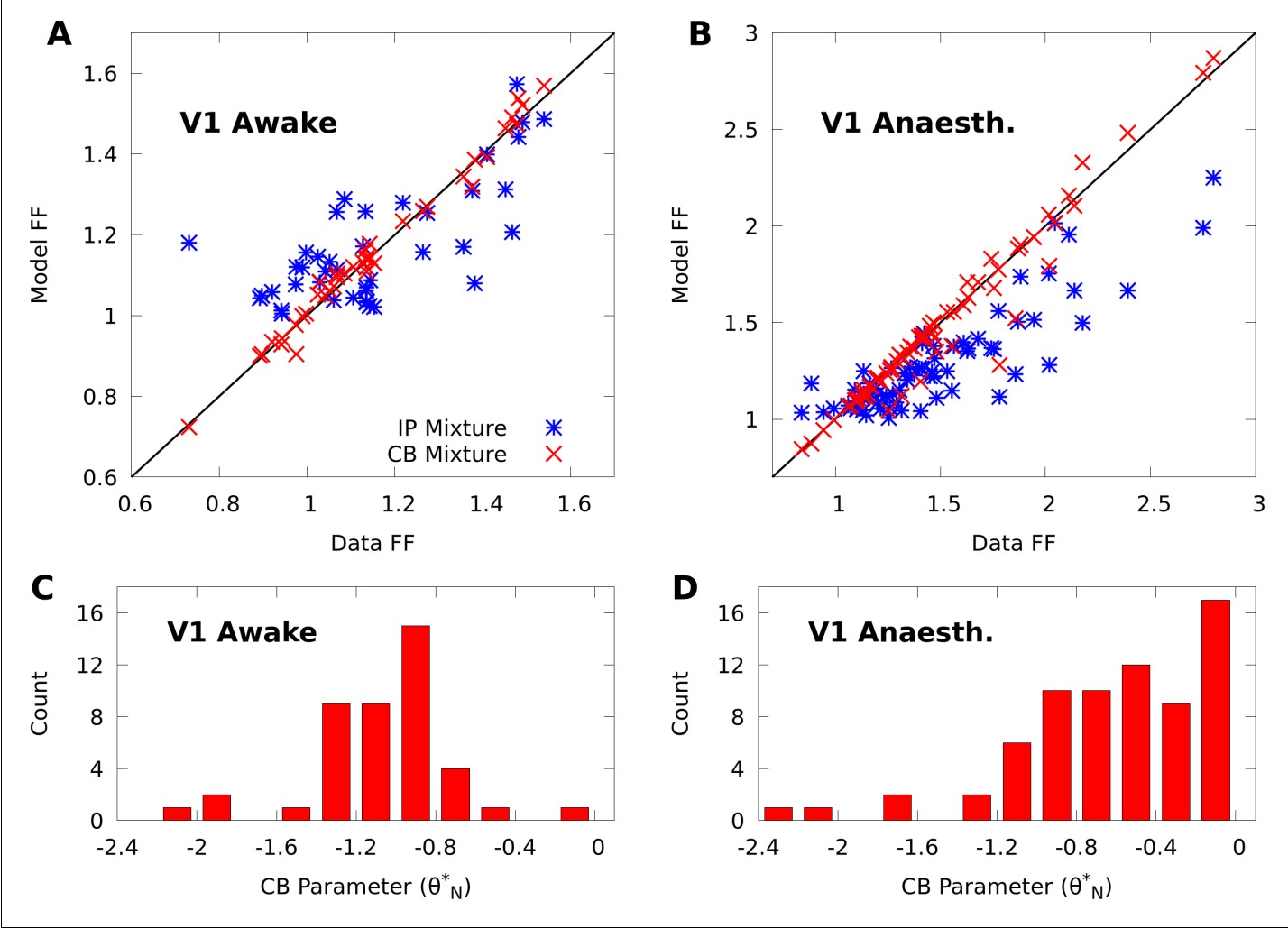

**Figure 2.** CoM-based parameters help Poisson mixtures capture individual variability in V1 responses to a single stimulus. We compare Independent Poisson (IP) mixtures (*Equation 1*) and CoM-Based (CB) mixtures (*Equation 2*) on neural population responses to stimulus orientation $x = 20°$ in V1 of awake ($d_N = 43$ neurons and $d_T = 355$ trials) and anaesthetized ($d_N = 70$ and $d_T = 1,200$) macaques; both mixtures are defined with $d_K = 5$ components for both data sets (see Materials and methods for training algorithms). A,B: Empirical Fano factors of the awake (A) and anaesthetized data (B), comparing IP (blue) and CB mixtures (red). C,D: Histogram of the CB parameters $\boldsymbol{\theta}_N^*$ for the CB mixture fits to the awake (C) and anaesthetized (D) data. Values of $\boldsymbol{\theta}_N^* < -1$ denote under-dispersed mixture components, values $> -1$ denote over-dispersed components.

$$p(\mathbf{n}, k \mid x) \propto e^{\boldsymbol{\theta}_N(x) \cdot \mathbf{n} + \boldsymbol{\theta}_N^* \cdot \mathbf{lf}(\mathbf{n}) + \boldsymbol{\theta}_K \cdot \boldsymbol{\delta}(k) + \mathbf{n} \cdot \boldsymbol{\Theta}_{NK} \cdot \boldsymbol{\delta}(k)}, \tag{3}$$

where $x$ is the stimulus, and $\boldsymbol{\theta}_N(x)$ are the stimulus-dependent baseline parameters, and we recover a minimal, IP-CM by setting $\boldsymbol{\theta}_N^* = -\mathbf{1}$.

The IP-CM again affords an intuitive interpretation: The first component of an IP-CM $p(\mathbf{n} \mid x, k = 1)$ has stimulus-dependent natural parameters $\boldsymbol{\theta}_N(x)$, and thus the stimulus-dependent firing rate, or tuning curve, of the $i$th neuron given $k = 1$ is $\mu_{i1}(x) = e^{\theta_{N,i}(x)}$, where $\theta_{N,i}(x)$ is the $i$th element of $\boldsymbol{\theta}_N(x)$. The natural parameters of the $k$th component for $k>1$ are then the sum of $\boldsymbol{\theta}_N(x)$ and column $k - 1$ of $\boldsymbol{\Theta}_{NK}$. As such, given $k>1$, the tuning curve of the $i$th neuron $\mu_{ik}(x) = \gamma_{i,(k-1)}\mu_{i1}(x)$ is a 'gain-modulated' version of $\mu_{i1}(x)$, where the gain $\gamma_{i,(k-1)}$ is the exponential function of element $i$ of column $k - 1$ of $\boldsymbol{\Theta}_{NK}$ (see *Equation 13*, Materials and methods). For a CB-CM this interpretation no longer holds exactly, but still serves as an approximate description of the behaviour of its components (see *Equation 16* and the accompanying discussions).

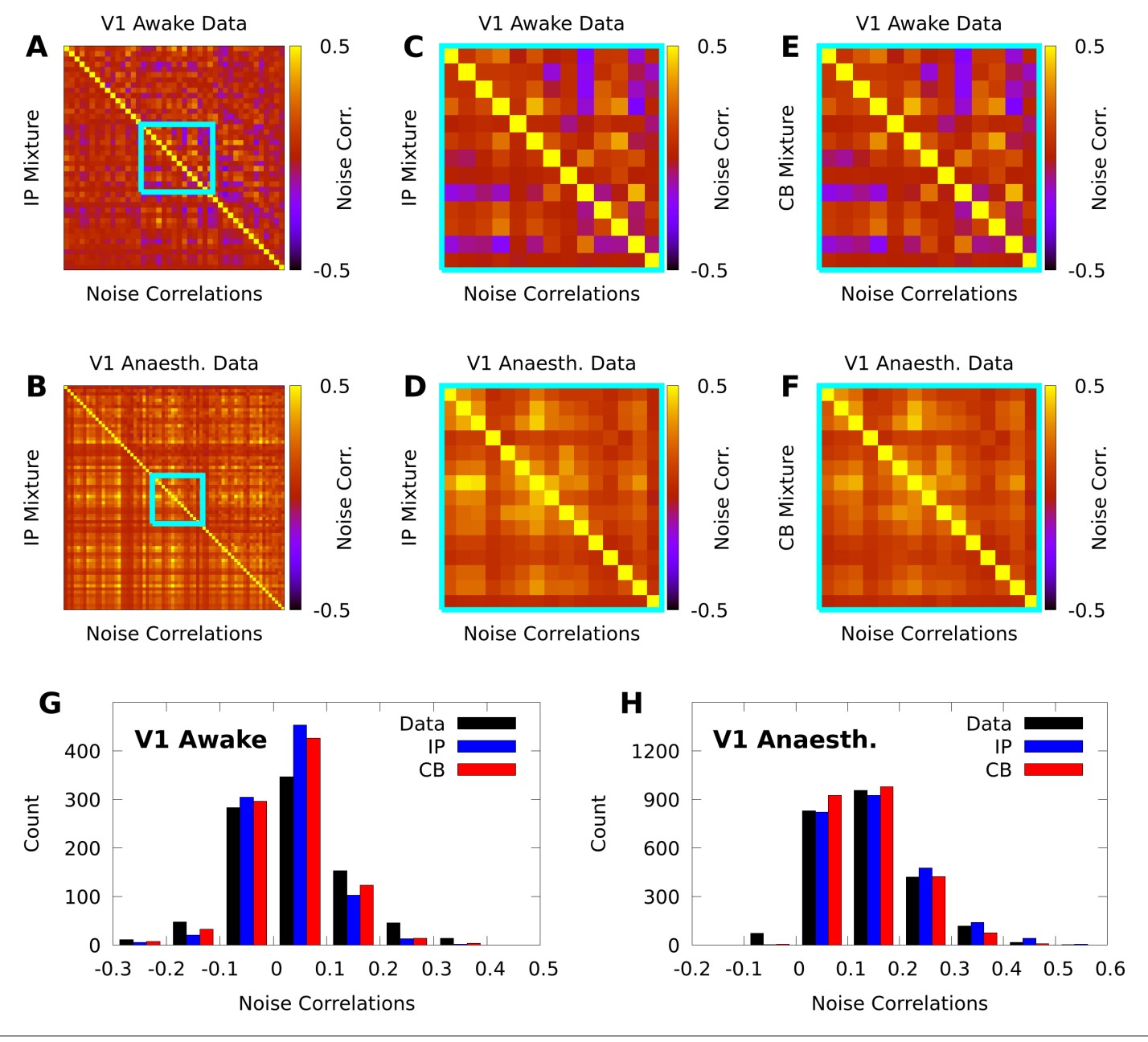

**Figure 3.** IP and CB mixtures effectively capture pairwise covariability in V1 responses to a single stimulus. Here we analyze the pairwise statistics of the same models from *Figure 2*. (A, B) Empirical correlation matrix (upper right triangles) of awake (A) and anaesthetized data (B), compared to the correlation matrix of the corresponding IP mixtures (lower left triangles). (C, D) Noise correlations highlighted in A and B, respectively. (E, F) Highlighted noise correlations for CB mixture fit. (G,H) Histogram of empirical noise correlations, and model correlations from IP and CB mixtures.

Towards understanding the expressive power of CMs, we study a minimal, CB-CM with $d_N = 20$ neurons, $d_K = 5$ mixture components, and randomly chosen parameters (see Materials and methods). Moreover, we assume that the stimulus is periodic (e.g. the orientation of a grating), and that the tuning curves of the component distributions $p(\mathbf{n} \mid x, k)$ have a von Mises shape, which is a widely applied model of neural tuning to periodic stimuli (*Herz et al., 2017*). We may achieve such a shape by defining the stimulus-dependent baseline parameters as $\boldsymbol{\theta}_N(x) = \boldsymbol{\theta}_N^0 + \boldsymbol{\Theta}_{NX} \cdot \mathbf{vm}(x)$, where $\boldsymbol{\theta}_N^0$ and $\boldsymbol{\Theta}_{NX}$ are parameters, and $\mathbf{vm}(x) = (\cos 2x, \sin 2x)$. *Figure 4A* shows that the tuning curves of the CB-CM neurons are approximately bell-shaped, yet many also exhibit significant deviations.

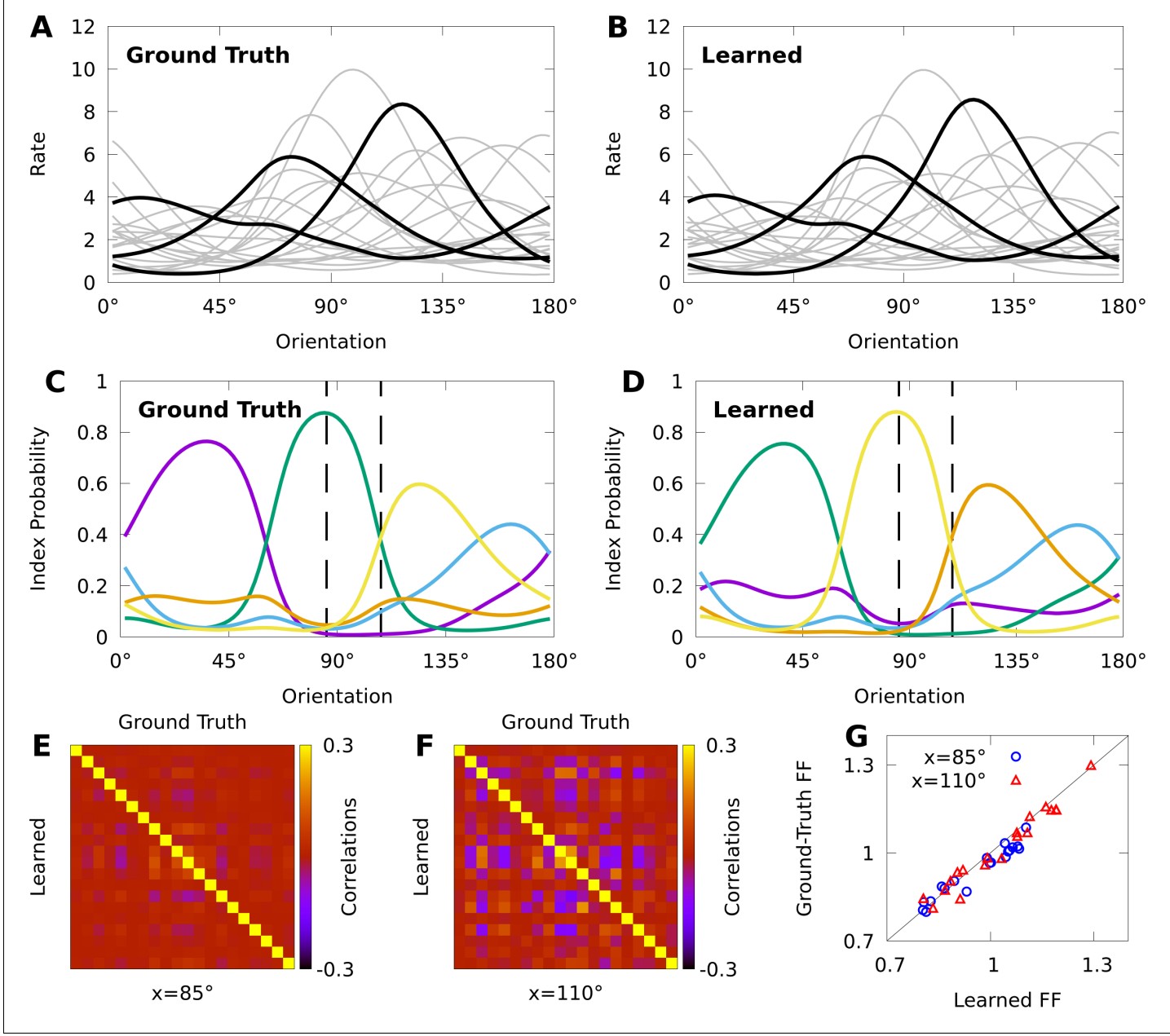

**Figure 4.** Expectation-maximization recovers a ground truth CoM-based, conditional mixture (CB-CM). We compare a ground truth, CB-CM with 20 neurons, five mixture components, von Mises-tuned components, and randomized parameters to a learned CB-CM fit to 2000 samples from the ground truth CB-CM. **A,B:** Tuning curves of the ground-truth CB-CM (**A**) and learned CB-CM (**B**). Three tuning curves are highlighted for effect. **C,D:** The orientation-dependent index probabilities of the ground truth CB-CM (**C**) and learned CB-CM (**D**), where colour indicates component index. Dashed lines indicate example stimulus-orientations used in **E**, **F**, and **G**. (**E**, **F**) The correlation matrix of the ground truth CB-CM (upper right), compared to the correlation matrix of the learned CB-CM (lower left) at stimulus orientations $x = 85°$ (**E**) and $x = 110°$ (**F**). (**G**) The FFs of the ground-truth CB-CM compared to the learned CB-CM at orientations $x = 85°$ (blue circles) and $x = 110°$ (red triangles).

We also study if CMs can be effectively fit to datasets comparable to those obtained in typical neurophysiology experiments. We generated 200 responses from the CB-CM described above — the ground truth CB-CM — to each of 10 orientations spread evenly over the half-circle, for a total of 2000 stimulus-response sample points. We then used this data to fit a CB-CM with the same number of components. Toward this aim, we derive an approximate EM algorithm to optimize model

parameters (see Materials and methods). *Figure 4B* shows that the tuning curves of the learned CB-CM are nearly indistinguishable from those of the ground truth CB-CM (*Figure 4B*, coefficient of determination $r^2 = 0.998$).

To reveal the orientation-dependent latent structure of the model, in *Figure 4C* we plot the index probability $p(k \mid x)$ for every $k$ as a function of the orientation $x$. In *Figure 4D* we show that the orientation-dependent index probabilities of the learned CB-CM qualitatively match the true index probabilities in *Figure 4C*. We also note that although the learned CB-CM does not correctly identify the indices themselves, this has no effect on the performance of the CB-CM.

The orientation-dependent index-probabilities provide a high-level picture of how the complexity and structure of model correlations varies with the orientation. The vertical dashed lines in *Figure 4C–D* denote two orientations that yield substantially different index probabilities $p(k \mid x)$. When a large number of index-probabilities are non-zero, the correlation-matrices of the CB-CM can exhibit complex correlations with both negative and positive values (*Figure 4E*). However, when one index dominates, the correlation structure largely disappears (*Figure 4F*). In *Figure 4G* we show that the FFs also depend on stimulus orientation. Lastly, we find that both the FF and the correlation-matrices of the learned CB-CM are nearly indistinguishable from the ground-truth CB-CM (*Figure 4E–G*).

In summary, our analyses show that minimal CB-CMs can express complex, stimulus-dependent response statistics, and that we can recover the structure of a ground truth CB-CM from realistic amounts of synthetic data with EM. In the following sections, we rigorously evaluate the performance of CMs on our awake and anaesthetized datasets.

## Conditional mixtures effectively model neural responses in macaque V1

A variety of models may be defined within the CM framework delineated by *Equations 1, 2 and 3*. Towards understanding how effectively CMs can model real data, we compare different variants by their cross-validated log-likelihood on both our awake and anaesthetized datasets; this is the same data used in *Figures 2* and *3* but now including all stimulus-conditions. We consider both IP and CB variants of each of the following conditional mixtures: (i) maximal CMs where we learn a distinct mixture for each of $d_X$ stimulus conditions, (ii) minimal CMs with von Mises-tuned components, and (iii) minimal CMs with *discrete*-tuned components given by $\boldsymbol{\theta}_N(x) = \boldsymbol{\theta}_N^0 + \boldsymbol{\Theta}_{NX} \cdot \boldsymbol{\delta}(x)$, where $\delta$ is the Kronecker delta vector with $d_X - 1$ elements, and $x$ is the index of the stimulus. In contrast with the von Mises CM, the discrete CM makes no assumptions about the form of component tuning. In *Table 1* we detail the number of parameters for all forms of CM.

To provide an interpretable measure of the relative performance of each CM variant, we define the 'information gain' as the difference between the estimated log-likelihood (base $e$) of the given CM and the log-likelihood of a von Mises-tuned, independent Poisson model, which is a standard model of uncorrelated neural responses to oriented stimuli (*Herz et al., 2017*). We then evaluate the predictive performance of our models with 10-fold cross-validation of the information gain.

*Table 2* shows that the CM variants considered achieve comparable performance, and perform substantially better than the independent Poisson lower bound on both the awake and anaesthetized data. *Figure 5* shows that a performance peak emerges smoothly as the model complexity (number of parameters) is increased. In all cases, the CB models outperform their IP counterparts, and typically with fewer parameters. The discrete CB-CMs achieve high performance on both datasets. In contrast, von Mises CMs perform well on the anaesthetized data but more poorly on the

---

**Table 1.** Parameter counts of CM models.

First row is number of parameters in IP models, second row is number of additional parameters in CB extensions of IP models, as a function of number of stimuli $d_S$, neurons $d_N$, and mixture components $d_K$.

**Model parameter formulae**

|  | Maximal | Von mises | Discrete |
|---|---|---|---|
| Num. Params | $d_S(d_N d_K + d_K - 1)$ | $(d_N + 1)(d_K - 1) + 3d_N$ | $(d_N + 1)(d_K - 1) + d_S d_N$ |
| Add. CB Params | $d_S d_N$ | $d_N$ | $d_N$ |

**Table 2.** Conditional mixtures models of neural responses in macaque V1 capture significant information about higher-order statistics.

We apply 10-fold cross-validation to estimate the mean and standard error of the information gain (model log-likelihood -log-likelihood of a non-mixed, independent Poisson model in nats/trial) on held-out data, from either awake (sample size $d_T = 3,168$, from $d_N = 43$ neurons, over $d_S = 9$ orientations) or anaesthetized ($d_T = 10,800$, $d_N = 70$, $d_S = 9$) macaque V1. We compare maximal CMs, minimal CMs with von Mises-tuned components, and minimal CMs with discrete-tuned components, and for each case we consider either IP or CB variants. For each variant, we indicate the number of CM components $d_K$ and the corresponding number of model parameters required to achieve peak information gain (cross-validated). For reference, the non-mixed, independent Poisson models use 129 and 210 parameters for the awake and anaesthetized data, respectively.

**Encoding performance**

| CM Variant | V1 awake data | | | V1 anaesthetized data | | |
|---|---|---|---|---|---|---|
| | Inf. Gain ($\frac{\text{Nats}}{\text{Trial}}$) | $d_K$ | # Params. | Inf. Gain ($\frac{\text{Nats}}{\text{Trial}}$) | $d_K$ | # Params. |
| Maximal IP | $2.30 \pm 0.32$ | 5 | 1971 | $8.77 \pm 0.71$ | 8 | 5103 |
| Maximal CB | $2.44 \pm 0.35$ | 5 | 2358 | $9.42 \pm 0.70$ | 7 | 5094 |
| Von Mises IP | $2.01 \pm 0.26$ | 45 | 2065 | $8.97 \pm 0.70$ | 40 | 2979 |
| Von Mises CB | $2.10 \pm 0.25$ | 40 | 1888 | $9.38 \pm 0.69$ | 35 | 2694 |
| Discrete IP | $2.25 \pm 0.28$ | 40 | 2103 | $9.17 \pm 0.70$ | 35 | 3044 |
| Discrete CB | $2.35 \pm 0.29$ | 30 | 1706 | $9.53 \pm 0.68$ | 30 | 2689 |
| Non-mixed IP | 0 | 1 | 129 | 0 | 1 | 210 |

awake data, and maximal CMs exhibit the opposite trend. Nevertheless, von Mises CMs solve a more difficult statistical problem as they also interpolate between stimulus conditions, and so may still prove relevant even where performance is limited. On the other hand, even though maximal CMs achieve high performance, they simply do so by replicating the high performance of stimulus-independent mixtures (*Figures 2* and *3*) at each stimulus condition, and require more parameters than minimal CMs to maximize performance.

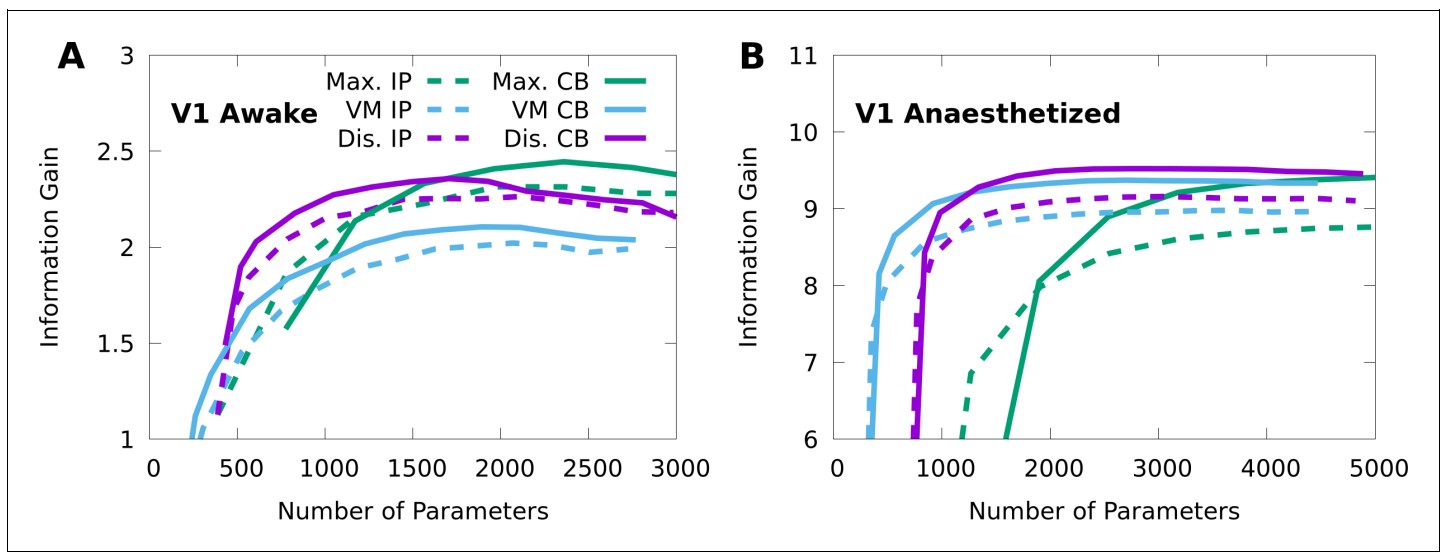

**Figure 5.** Finding the optimal number of parameters for CMs to model neural responses in macaque V1. 10-fold cross-validation of the information gain given awake V1 data (**A**) and anaesthetized V1 data (**B**), as a function of the number of model parameters, for multiple forms of CM: maximal CMs (green); minimal CMs with von Mises component tuning (blue); minimal CMs with discrete component tuning (purple); and for each case we consider either IP (dashed lines) or CB (solid lines) variants. Standard errors of the information gain are not depicted to avoid visual clutter, however they are approximately independent of the number of model parameters, and match the values indicated in *Table 2*.

# Conditional mixtures facilitate accurate and efficient decoding of neural responses

To demonstrate that CMs model the neural code, we must show that CMs not only capture the features of neural responses, but that these features also encode stimulus-information. Given an encoding model $p(\mathbf{n} \mid x)$ and a response from the model $\mathbf{n}$, we may optimally decode the information in the response about the stimulus $x$ by applying Bayes' rule $p(x \mid \mathbf{n}) \propto p(\mathbf{n} \mid x)p(x)$, where $p(x \mid \mathbf{n})$ is the posterior distribution (the decoded information), and $p(x)$ represents our prior assumptions about the stimulus (*Zemel et al., 1998*). When we do not know the true encoding model, and rather fit a statistical model to stimulus-response data, using the statistical model for Bayesian decoding and analyzing its performance can tell us how well it captures the features of the neural code.

We analyze the performance of Bayesian decoders based on CMs by quantifying their decoding performance, and comparing the results to other common approaches to decoding. We evaluate decoding performance with the 10-fold cross-validation log-posterior probability $\log p(x \mid \mathbf{n})$ (base $e$) of the true stimulus value $x$, for both our awake and anaesthetized V1 datasets. With regard to choosing the number of components $d_K$, we analyze the decoding performance of CMs that achieved the best *encoding* performance based as indicated in *Table 2* and depicted *Figure 5*. We do this to demonstrate how well a single model can simultaneously perform at both encoding and decoding, instead of applying distinct procedures for selecting CMs based on decoding performance (see Materials and methods for a summary of trade-offs when choosing $d_K$).

In our comparisons we focus on minimal, discrete CMs as overall they achieved high performance on both datasets (*Figure 5*). To characterize the importance of neural correlations to Bayesian decoding, we compare our CMs to the decoding performance of independent Poisson models with discrete tuning (Non-mixed IP). To characterize the optimality of our Bayesian decoders, we also evaluate the performance of linear multiclass decoders (Linear), as well nonlinear multiclass decoders defined as artificial neural networks (ANNs) with two hidden layers and a cross-validated number of hidden units (for details on the training and model selection procedure, see Materials and methods).

*Table 3* shows that on the awake data, the performance of the CMs is statistically indistinguishable from the ANN, and the CMs and the ANN significantly exceed the performance of both the Linear and Non-mixed IP models. On the anaesthetized data, the minimal CM approaches the performance of the ANN, and the minimal CMs and ANN models again exceed the performance of the Non-mixed IP and Linear models. Yet in this case, the Linear model is much more competitive, whereas the Non-mixed IP model performs very poorly, possibly because of the larger magnitude of noise correlations in this data. In Appendix 2, we also report that a Bayesian decoder based on a factor analysis (FA) encoding model performed inconsistently, and poorly relative to CMs, as it would

**Table 3.** CMs support high-performance decoding of neural responses in macaque V1.
We apply 10-fold cross-validation to estimate the mean and standard error of the average log-posteriors $\log p(x \mid \mathbf{n})$ on held-out data, from either awake or anaesthetized macaque V1. We compare discrete, minimal, CB-CM (CB-CM) and IP-CM (IP-CM); an independent Poisson model with discrete tuning (Non-mixed IP); a multiclass linear decoder (Linear); and a multiclass nonlinear decoder defined as an artificial neural network with two hidden layers (Artificial NN). The number of CM components $d_K$ was chosen to achieve peak information gain in *Figure 5*. The number of ANN hidden units was chosen based on peak cross-validation performance. In all cases we also indicate the number of model parameters required to achieve the indicated performance.

**Decoding performance**

|  | V1 awake data | | V1 anaesthetized data | |
| --- | --- | --- | --- | --- |
|  | Average Log-Post. | Num. Params. | Average Log-Post. | Num. Params. |
| IP-CM | $-0.207 \pm 0.039$ | 2103 | $-0.448 \pm 0.026$ | 3044 |
| CB-CM | $-0.206 \pm 0.043$ | 1706 | $-0.441 \pm 0.023$ | 2689 |
| Non-mixed IP | $-0.272 \pm 0.067$ | 387 | $-0.967 \pm 0.071$ | 630 |
| Linear | $-0.256 \pm 0.053$ | 352 | $-0.457 \pm 0.019$ | 568 |
| Artificial NN | $-0.200 \pm 0.032$ | 527,108 | $-0.426 \pm 0.015$ | 408,008 |

occasionally assign numerically 0 probability to the true stimulus, and thus score an average log-posterior of negative infinity. In Appendix 2, we present preliminary evidence that this is because CMs capture higher order structure that FA cannot.

On both the awake and anaesthetized data the ANN requires two orders of magnitude more parameters than the CMs to achieve its performance gains. In addition, the CB-CM achieves marginally better performance with fewer parameters than the IP-CM, indicating that although modelling individual variability is not essential for effective Bayesian decoding, doing so still results in a more parsimonious model of the neural code. In Appendix 3, we report a sample complexity analysis of CM encoding and decoding performance. We found that whereas our anaesthetized V1 dataset (sample size $d_T = 10,800$) was large enough to saturate the performance of our models, a larger awake V1 dataset ($d_T = 3,168$) could yield further improvements to decoding performance.

We also consider widely used alternative measures of decoding performance, namely the Fisher information (FI), which is an upper bound on the average precision (inverse variance) of the posterior (*Brunel and Nadal, 1998*), as well as the linear Fisher information (LFI), which is a linear approximation of the FI (*Seriès et al., 2004*) corresponding to the accuracy of the optimal, unbiased linear decoder of the stimulus (*Kanitscheider et al., 2015a*). The FI is especially helpful when the posterior cannot be evaluated directly (such as when it is continuous), and is widely adopted in theoretical (*Abbott and Dayan, 1999*; *Beck et al., 2011b*; *Ecker et al., 2014*; *Moreno-Bote et al., 2014*; *Kohn et al., 2016*) and experimental (*Ecker et al., 2011*; *Kafashan et al., 2021*; *Rumyantsev et al., 2020*) studies of neural coding. As with other models based on exponential family theory (*Ma et al., 2006*; *Beck et al., 2011b*; *Ecker et al., 2016*), the FI of a minimal CM may be expressed in closed-form, and is equal to its LFI (see Materials and methods), and therefore minimal CMs can be used to study FI analytically and obtain model-based estimates of FI from data.

To study how well CMs capture FI, we defined 40 random subpopulations of $d_N = 20$ neurons from both our V1 datasets, fit von Mises IP-CMs to the responses of each subpopulation, and used these learned models as ground-truth populations. We then generated 50 responses at each of 10 evenly spaced orientations from each ground truth IP-CM, for a total of $d_T = 500$ responses per ground-truth model. We then fit a new IP-CM to each set of 500 responses, and compared the FI of the re-fit CM to the FI of the ground-truth CM at 50 evenly spaced orientations. Pooled over all populations and orientations, the relative error of the estimated FI was $-12.8\% \pm 18.6\%$ on the awake data and $-9.1\% \pm 22.4\%$ on the anaesthetized data, suggesting that IP-CMs can recover and even interpolate approximate FIs of ground-truth populations from modest amounts of data.

To summarize, CMs support accurate Bayesian decoding in awake and anaesthetized macaque V1 recordings, and are competitive with nonlinear decoders with two orders of magnitude more parameters. Moreover, CMs afford closed-form expressions of FI and can interpolate good estimates of FI from modest amounts of data, and thereby support analyses of neural data based on this widely applied theoretical tool.

## Constrained conditional mixtures support linear probabilistic population coding

Having shown that minimal CMs can both capture the statistics of neural encoding and facilitate accurate Bayesian decoding, we now aim to show how they relate to an influential theory of neural coding known as probabilistic population codes (PPCs), which describes how neural circuits process information in terms of encoding and Bayesian decoding (*Zemel et al., 1998*). In particular, linear probabilistic population codes (LPPCs) are PPCs with a restricted encoding model, that explain numerous features of neural coding in the brain (*Ma et al., 2006*; *Beck et al., 2008*; *Beck et al., 2011a*).

In general, an exponential family of distributions that depend on some stimulus $x$ may be expressed as $p(\mathbf{n} \mid x) = e^{\boldsymbol{\theta}_N(x) \cdot \mathbf{s}_N(\mathbf{n}) - \psi_N(\boldsymbol{\theta}_N(x))} \mu(\mathbf{n})$, where $\mathbf{s}_N$ is the sufficient statistic, $\mu$ is the base measure, and $\psi_N(\boldsymbol{\theta}_N(x))$ is known as the log-partition function (in *Equations 1-3* we used the proportionality symbol $\propto$ to avoid writing the log-partition functions explicitly). A PPC is an LPPC when its encoding model is in the so-called exponential family with linear sufficient statistics (EFLSS), which has the form $p(\mathbf{n} \mid x) = e^{\boldsymbol{\theta}_N(x) \cdot \mathbf{n}} \phi(\mathbf{n})$ for some functions $\phi(\mathbf{n})$ and $\boldsymbol{\theta}_N(x)$ (*Beck et al., 2011a*). If we equate the two expressions $e^{\boldsymbol{\theta}_N(x) \cdot \mathbf{s}_N(\mathbf{n}) - \psi_N(\boldsymbol{\theta}_N(x))} \mu(\mathbf{n}) = e^{\boldsymbol{\theta}_N(x) \cdot \mathbf{n}} \phi(\mathbf{n})$ we see that an EFLSS is a stimulus-dependent exponential family that satisfies two constraints: that the sufficient statistic $\mathbf{s}_N(\mathbf{n}) = \mathbf{n}$ is

linear, and that the log-partition function $\psi_N(\boldsymbol{\theta}_N(x)) = \alpha$ does not depend on the stimulus, so that $\phi(\mathbf{n}) = e^{-\alpha}\mu(\mathbf{n})$.

As presented, the EFLSS is a mathematical model that does not have fittable parameters. We wish to express CMs as a form of EFLSS in order to show how a fittable model could be compatible with LPPC theory. If we return to the general expression for a minimal CM (*Equation 3*) and assume that the log-partition function is given by the constant $\alpha$, then we may write

$$p(\mathbf{n} \mid x) = \sum_k p(\mathbf{n}, k \mid x) = e^{\boldsymbol{\theta}_N(x)\cdot\mathbf{n}}\left(e^{\boldsymbol{\theta}_N^*\cdot\mathbf{lf}(\mathbf{n})-\alpha}\sum_k e^{\boldsymbol{\theta}_K\cdot\boldsymbol{\delta}(k)+\mathbf{n}\cdot\boldsymbol{\Theta}_{NK}\cdot\boldsymbol{\delta}(k)}\right) = e^{\boldsymbol{\theta}_N(x)\cdot\mathbf{n}}\phi(\mathbf{n}), \tag{4}$$

where $\phi(\mathbf{n}) = e^{\boldsymbol{\theta}_N^*\cdot\mathbf{lf}(\mathbf{n})-\alpha}\sum_k e^{\boldsymbol{\theta}_K\cdot\boldsymbol{\delta}(k)+\mathbf{n}\cdot\boldsymbol{\Theta}_{NK}\cdot\boldsymbol{\delta}(k)}$, such that the given CM is in the EFLSS. Observe that this equation only holds due to the specific structure of minimal CMs: if the parameters $\boldsymbol{\theta}_N^*$, $\boldsymbol{\theta}_K$, or $\boldsymbol{\Theta}_{NK}$ would depend on the stimulus, then it would not be possible to absorb them into the function $\phi(\mathbf{n})$.

Ultimately, this equivalence between constrained CMs and EFLSSs allows LPPC theory to be applied to constrained CMs, and provides theorists working on neural coding with an effective statistical tool that can help validate their hypotheses.

## Minimal conditional mixtures capture information-limiting correlations

Our last aim is to demonstrate that CMs can approximately represent a central phenomenon in neural coding known as information-limiting correlations, which are neural correlations that fundamentally limit stimulus-information in neural circuits (*Moreno-Bote et al., 2014*; *Montijn et al., 2019*; *Bartolo et al., 2020*; *Kafashan et al., 2021*; *Rumyantsev et al., 2020*). To illustrate this, we generate population responses with limited information, and then fit an IP-CM to these responses and study the learned latent representation. In particular, we consider a source population of 200 independent Poisson neurons $p(\mathbf{n} \mid s)$ with homogeneous, von Mises tuning curves responding to a noisy stimulus-orientation $s$, where the noise $p(s \mid x)$ follows a von Mises distribution centred at the true stimulus-orientation $x$ (see Materials and methods). In *Figure 6A* we show that, as expected, the average FI in the source population about the noisy orientation $s$ grows linearly with the size of randomized subpopulations, although the FI about the true orientation $x$ is theoretically bounded by the precision (inverse variance) of the sensory noise.

Even though the neurons in the source model are uncorrelated, sensory noise ensures that the encoding model $p(\mathbf{n} \mid x) = \int p(\mathbf{n} \mid s)p(s \mid x)ds$ contains information-limiting correlations that bound the FI about $x$(*Moreno-Bote et al., 2014*; *Kanitscheider et al., 2015b*). Information-limiting correlations can be small and difficult to capture, and to understand how CMs learn in the presence of information-limiting noise correlations, we fit a von Mises IP-CM $q(\mathbf{n} \mid x)$ with $d_K = 20$ mixture components to $d_T = 10,000$ responses from the information-limited model $p(\mathbf{n} \mid x)$. *Figure 6A* (purple) shows that the FI of the learned CM $q(\mathbf{n} \mid x)$ appears to saturate near the precision of the sensory noise, indicating that the learned CM approximates the information-limiting correlations present in $p(\mathbf{n} \mid x)$.

To understand how the learned CM approximates these information-limiting correlations, we study the relation between the latent structure of the model and how it generates population activity. For an IP-CM, the orientation-dependent index-probabilities may be expressed as $p(k \mid x) \propto e^{\boldsymbol{\theta}_K\cdot\boldsymbol{\delta}(k)+\sum_{i=1}^{d_N}\mu_{ik}(x)}$, where $\mu_{ik}(x)$ is the tuning curve of the $i$ th neuron under component $k$. In *Figure 6B*, we plot the sum of the tuning curves $\sum_{i=1}^{d_N}\mu_{ik}(x)$ for each component $k$ as a function of orientation, and we see that each component concentrates the tuning of the population around a particular orientation. This encourages the probability of each component to also concentrate around a particular orientation, and in *Figure 6C* we see that, given the true orientation $x = 90°$, there are three components with probabilities substantially greater than 0.

Because there are essentially three components that are relevant to the responses of the IP-CM to the true orientation $x = 90°$, generating a response from the CM approximately reduces to generating a response from one of the three possible component IP distributions. In *Figure 6D–F*, we depict a response to $x = 90°$ from each of the three component IP distributions, as well as the optimal posterior based on the learned IP-CM (purple lines), and a suboptimal posterior based on the source model (i.e. ignoring noise correlations; green lines). We observe that the trial-to-trial variability of the learned IP-CM results in random shifts of the peak neural activity away from the true orientation, thus limiting information. Furthermore, when the response of the population is concentrated

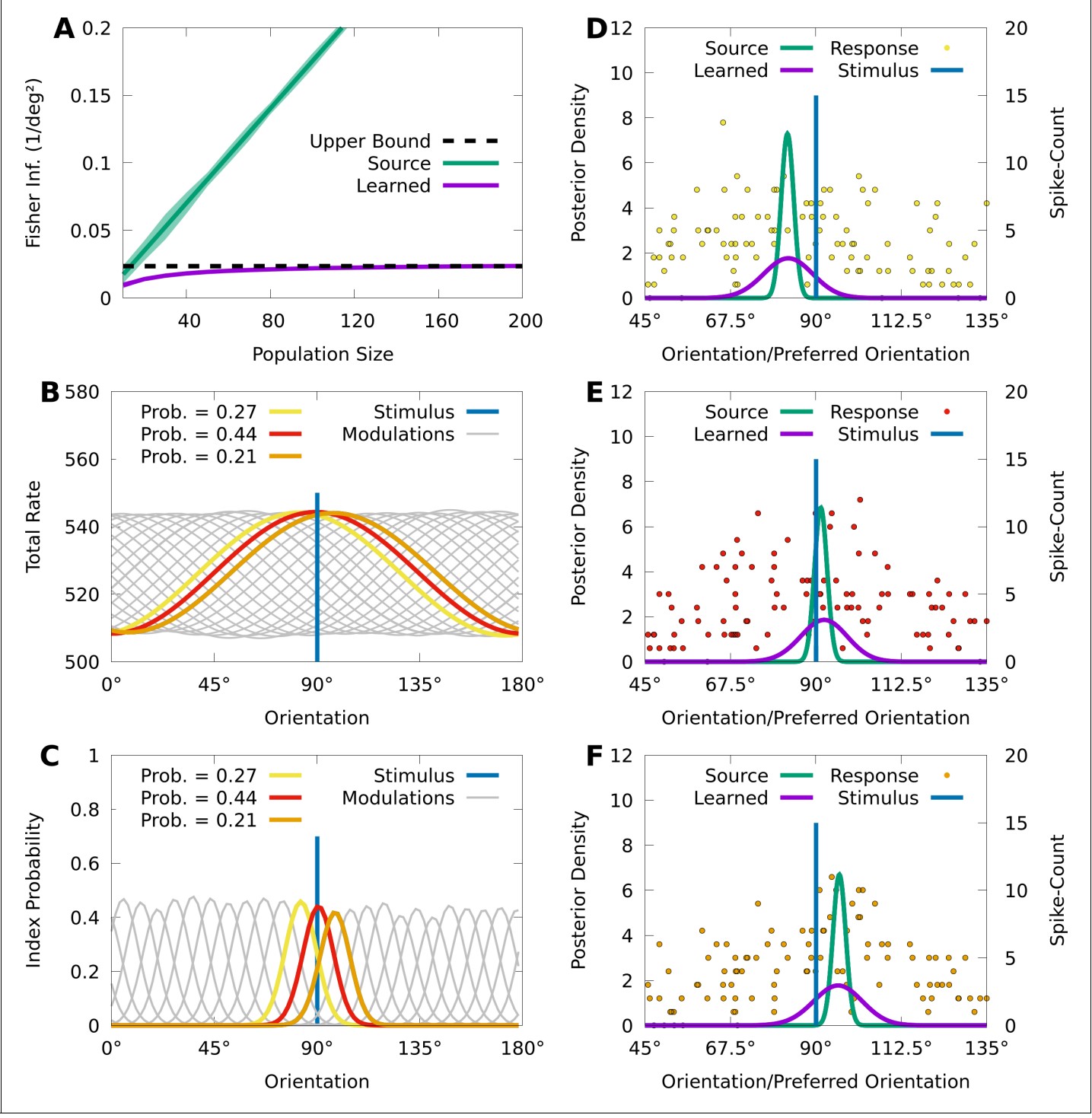

**Figure 6.** CMs can capture information-limiting correlations in data. We consider a von Mises-tuned, independent Poisson source model (green) with $d_K = 200$ neurons, and an information-limited, IP-CM (purple) with $d_K = 25$ components, fit to 10,000 responses of the source-model to stimuli obscured by von Mises noise. In B-F we consider a stimulus-orientation $x = 90°$ (blue line). (A) The average (lines) and standard deviation (filled area) of the FI over orientations, for the source (green) and information-limited (purple) models, as a function of random subpopulations, starting with ten neurons, and gradually reintroducing missing neurons. Dashed black line indicates the theoretical upper bound. (B) The sum of the firing rates of the modulated IP-CM for all indices $k>1$ (lines) as a function of orientation, with three modulated IP-CMs highlighted (red, yellow, and orange lines) corresponding to the highlighted indices in C. (C) The index-probability curves (lines) of the IP-CM for indices $k>1$ and the intersection (red, yellow, and orange circles) of the stimulus with three curves (orange, yellow, and orange lines). (D-F) Three responses from the yellow (D; yellow points), red (E; red points), and

*Figure 6 continued on next page*

*Figure 6 continued*

orange modulated IP-CMs (**F**; orange points) indicated in **C**. For each response we plot the posterior based on the source model (green line) and the information-limited model (purple line).

at the true orientation (*Figure 6E*), the suboptimal posterior assigns a high probability to the true orientation, whereas when the responses are biased away from the true orientation (*Figure 6D and F*) the suboptimal posterior assigns nearly 0 probability to the true orientation. This is in contrast to the optimal posterior, which always assigns a significant probability to the true orientation.

In summary, CMs can effectively approximate information-limiting correlations, and the simple latent structure of CMs could help reveal the presence of information-limiting correlations in data.

## Discussion

We introduced a latent variable exponential family formulation of Poisson mixtures. We showed how this formulation allows us to effectively extend Poisson mixtures both to capture sub-Poisson variability, and to incorporate stimulus dependence using conditional mixtures. Our analyses and simulations showed that these conditional mixtures (CMs) can be fit efficiently and recover ground truth models in synthetic data, capture a wide range of V1 response statistics in real data, and can be easily inverted to obtain accurate Bayesian decoding that is competitive with nonlinear decoders, while using orders of magnitude less parameters. In addition, we illustrated how the latent structure of CMs can represent a fundamental feature of the neural code, namely information-limiting correlations.

Our framework is particularly relevant for probabilistic theories of neural coding based on the theory of exponential families (*Beck et al., 2007*), which include theories that address the linearity of Bayesian inference in neural circuits (*Ma et al., 2006*), the role of phenomena such as divisive normalization in neural computation (*Beck et al., 2011a*), Bayesian inference about dynamic stimuli (*Makin et al., 2015*; *Sokoloski, 2017*), and the metabolic efficiency of neural coding (*Ganguli and Simoncelli, 2014*; *Yerxa et al., 2020*). These theories have proven difficult to validate quantitatively with neural data due to a lack of statistical models which are both compatible with their exponential family formulation (see *Equation 4*), and can model correlated activity in recordings of large neural populations. Our work suggests that CMs can overcome these difficulties, and help connect this rich mathematical theory of neural coding with the state-of-the-art in parallel recording technologies.

CMs are not limited to modelling neural responses to stimuli and can model how arbitrary experimental variables modulate neural variability and covariability. Examples of experimental variables that have measurable effects on neural covariability include the spatial and temporal context around a stimulus (*Snyder et al., 2014*; *Snow et al., 2016*; *Snow et al., 2017*; *Festa et al., 2020*), as well as task-variables and the attentional state of the animal (*Cohen and Maunsell, 2009*; *Mitchell et al., 2009*; *Ruff and Cohen, 2014*; *Maunsell, 2015*; *Rabinowitz et al., 2015*; *Verhoef and Maunsell, 2017*; *Bondy et al., 2018*). Each of these variables could be incorporated into a CM by either replacing the stimulus-variable in our equations with the variable of interest, or combining it with the stimulus-variable to construct a CM with multivariate dependence. This would allow researchers to explore how the stimulus and the experimental variables mutually interact to shape variability and covariability in large populations of neurons.

To understand how this variability and covariability effects neural coding, latent variable models such as CMs are often applied to extract interpretable features of the neural code from data (*Whiteway and Butts, 2019*). The latent states of a CM provide a soft classification of neural activity, and we may apply CMs to model how an experimental variable modulates the class membership of population activity. In the studies on experimental variables listed above, models of neural activity yielded predictions of perceptual and behavioural performance. Because CMs support Bayesian decoding, a CM can also make predictions about how a particular class of neurons is likely to modulate perception and behaviour, and we may then test these predictions with experimental interventions on the neurons themselves (*Panzeri et al., 2017*). In this manner, we believe CMs could form a critical part of a rigorous, Bayesian framework for 'cracking the neural code' in large populations of neurons.

Outside of the framework of mixture models, there are broader possibilities for designing conditional, latent-variable models which have the minimal, exponential family structure of *Equation 3*, yet for which the latent variable is not a finite index. We make use of finite mixture models in this paper primarily because mixture models are analytically tractable, even when mixing Poisson distributions. In contrast, models with Gaussian latent variables are analytically tractable when the observations are also Gaussian, but not in general. Nevertheless, if the relevant formulae and computations can be effectively approximated, then many of the advantages of CMs could be preserved even when using continuous latent variables. For example, if the expectation step in our EM algorithm does not have a closed-form expression, it might be possible to approximate it with contrastive divergence (*Hinton, 2002*).

In our applications, we considered one-dimensional stimuli and implemented the stimulus-dependence of the CM parameters with linearly parameterized functions. Nevertheless, this stimulus dependence can be implemented by arbitrary parametric functions of high-dimensional variables such as deep neural networks, and CMs can also incorporate history-dependence via recurrent connectivity (see Appendix 4). As such, CMs have the potential to integrate encoding models of higher cortical areas (*Yamins et al., 2014*) with models of the temporal features of the neural code (*Pillow et al., 2008*; *Park et al., 2014*; *Runyan et al., 2017*), towards analyzing the neural code in dynamic, correlated neural populations in higher cortex. Finally, outside of neuroscience, high-dimensional count data exists in many fields such as corpus linguistics and genomics (*Inouye et al., 2017*), and researchers who aim to understand how this data depends on history or additional variables could benefit from our techniques.

## Materials and methods

### Notation

We use capital, bold letters (e.g. $\boldsymbol{\Theta}$) to indicate matrices; small, bold letters (e.g. $\boldsymbol{\theta}$) to indicate vectors; and regular letters (e.g. $\theta$) to indicate scalars. We use subscript capital letters to indicate the role of a given variable, so that, in *Equation 1* for example, $\boldsymbol{\theta}_K$ are the natural parameters that bias the index-probabilities, $\boldsymbol{\theta}_N$ are the baseline natural parameters of the neural firing rates, and $\boldsymbol{\Theta}_{NK}$ is the matrix of parameters through which the indices and rates interact.

We denote the $i$ th element of a vector $\boldsymbol{\theta}$ by $\theta_i$, or e.g. of the vector $\boldsymbol{\theta}_K$ by $\theta_{K,i}$. We denote the $i$ th row or $j$ th column of $\boldsymbol{\Theta}$ by $\boldsymbol{\theta}_i$ or $\boldsymbol{\theta}_j$, respectively, and always state whether we are considering a row or column of the given matrix. When referring to the $j$ th element of a vector $\boldsymbol{\theta}_i$ indexed by $i$, we write $\theta_{ij}$. Finally, when indexing data points from a sample, or parameters that are tied to individual data points, we use parenthesized, superscript letters, e.g. $x^{(i)}$, or $\boldsymbol{\theta}_N^{(i)}$.

### Poisson mixtures and their moments

The following derivations were presented in a more general form in *Karlis and Meligkotsidou, 2007*, but we present the simpler case here for completeness. A Poisson distribution has the form $p(\boldsymbol{n}; \boldsymbol{\lambda}) = \frac{\lambda^n e^{-\lambda}}{n!}$, where $n$ is the count and $\lambda$ is the rate (in our case, spike count and firing rate, respectively). We may use a Poisson model to define a distribution over $d_N$ spike counts $\boldsymbol{n} = (n_1, \ldots, n_{d_N})$ by supposing that the neurons generate spikes independently of one another, leading to the independent Poisson model $p(\boldsymbol{n}; \boldsymbol{\lambda}) = \prod_{i=1}^{d_N} p(n_i; \lambda_i)$ with firing rates $\boldsymbol{\lambda} = (\lambda_1, \ldots, \lambda_{d_N})$. Finally, if we consider the $d_K$ rate vectors $\boldsymbol{\lambda}_1, \ldots, \boldsymbol{\lambda}_{d_K}$, and $d_K$ weights $w_1, \ldots, w_{d_K}$, where $0 \leq w_k$ for all $k$, and $w_1 = 1 - \sum_{k=2}^{d_K} w_k$, we then define a mixture of Poisson distributions as a latent variable model $p(\boldsymbol{n}) = \sum_k \boldsymbol{p}(\boldsymbol{n} \mid \boldsymbol{k}) \boldsymbol{p}(\boldsymbol{k}) = \sum_k \boldsymbol{p}(\boldsymbol{n}, \boldsymbol{k})$, where $p(\boldsymbol{n} \mid \boldsymbol{k}) = \boldsymbol{p}(\boldsymbol{n}; \boldsymbol{\lambda}_k)$, and $p(k) = w_k$.

The mean $\mu_i$ of the $i$ th neuron of a mixture of independent Poisson distributions is

$$\mu_i = \sum_{n_i=0}^{\infty} \sum_{k=1}^{d_K} p(n_i \mid k) p(k) n_i = \sum_{k=1}^{d_K} p(k) \sum_{n_i=0}^{\infty} p(n_i \mid k) n_i = \sum_{k=1}^{d_K} w_k \lambda_{ik}. \tag{5}$$

The variance $\sigma_i^2$ of neuron $i$ is

$$\sigma_i^2 = \sum_{n_i=0}^{\infty} p(n_i) n_i^2 - \mu_i^2 = \sum_{k=1}^{d_K} p(k) \sum_{n_i=0}^{\infty} p(n_i \mid k) n_i^2 - \mu_i^2$$

$$= \sum_{k=1}^{d_K} p(k)(\sigma_{ik}^2 + \lambda_{ik}^2) - \mu_i^2 = \mu_i + \sum_{k=1}^{d_K} w_k(\lambda_{ik} - \mu_i)^2, \tag{6}$$

where $\sigma_{ik}^2 = \lambda_{ik}$ is the variance of the $i$ th neuron under the $k$ th component distribution, that is the variance of $p(n_i \mid k)$, and where $\sum_{n_i=0}^{\infty} p(n_i \mid k) n_i^2 = \sigma_{ik}^2 + \lambda_{ik}^2$, and $\sum_{k=1}^{d_K} w_k \lambda_{ik}^2 - \mu_i^2 = \sum_{k=1}^{d_K} w_k(\lambda_{ik} - \mu_i)^2$ both follow from the fact that a distribution's variance equals the difference between its second moment and squared first moment.

The covariance $\sigma_{ij}^2$ between spike-counts $n_i$ and $n_j$ for $i \neq j$ is then

$$\sigma_{ij}^2 = \sum_{n_i=0}^{\infty} \sum_{n_j=0}^{\infty} p(n_i, n_j)(n_i - \mu_i)(n_j - \mu_j) = \sum_{k=1}^{d_K} p(k) \sum_{n_i=0}^{\infty} \sum_{n_j=0}^{\infty} p(n_i, n_j \mid k)(n_i - \mu_i)(n_j - \mu_j)$$

$$= \sum_{k=1}^{d_K} p(k) \sum_{n_i=0}^{\infty} p(n_i \mid k)(n_i - \mu_i) \sum_{n_j=0}^{\infty} p(n_j \mid k)(n_j - \mu_j) = \sum_{k=1}^{d_K} w_k(\lambda_{ik} - \mu_i)(\lambda_{jk} - \mu_j). \tag{7}$$

Observe that if $w_k = \frac{1}{d_K-1}$, then $\sigma_{ij}^2$ is simply the sample covariance between $i$ and $j$, where the sample is composed of the rate components of the $i$ th and $j$ th neurons. *Equation 7* thus implies that Poisson mixtures can model arbitrary covariances. Nevertheless, *Equation 6* shows that the variance of individual neurons is restricted to being larger than their means.

## Exponential family mixture models

In this section, we show that the latent variable form for Poisson mixtures we introduced above is a member of the class of models known as exponential families. An exponential family distribution $p(x)$ over some data $x$ has the form $p(x) = e^{\boldsymbol{\theta} \cdot \mathbf{s}(x) - \psi(\boldsymbol{\theta})} b(x)$, where $\boldsymbol{\theta}$ are the so-called natural parameters, $\mathbf{s}(x)$ is a vector-valued function of the data called the sufficient statistic, $b(x)$ is a scalar-valued function called the base measure, and $\psi(\boldsymbol{\theta}) = \log \int e^{\boldsymbol{\theta} \cdot \mathbf{s}(x)} b(x) dx$ is the log-partition function (*Wainwright and Jordan, 2008*). In the context of Poisson mixture models, we note that an independent Poisson model $p(\mathbf{n}; \boldsymbol{\lambda})$ is an exponential family, with natural parameters $\boldsymbol{\theta}_N$ given by $\theta_{N,i} = \log \lambda_i$, base measure $b(\boldsymbol{n}) = \frac{1}{\prod_i n!}$ and sufficient statistic $\mathbf{s}_N(\boldsymbol{n}) = \boldsymbol{n}$, and log-partition function $\psi_N(\boldsymbol{\theta}_N) = \sum_{i=1}^{d_N} e^{\theta_{N,i}}$. Moreover, the distribution of component indices $p(k) = w_k$ (also known as a categorical distribution) also has an exponential family form, with natural parameters $\theta_{K,k} = \log \frac{w_{k+1}}{w_1}$ for $1 \leq k < d_K$, sufficient statistic $\boldsymbol{\delta}(k) = (\delta_2(k), \ldots, \delta_{d_K}(k))$, base measure $b(k) = 1$, and log-partition function $\psi_K(\boldsymbol{\theta}_K) = \log(1 + \sum_{k=1}^{d_K-1} e^{\theta_{K,k}})$. Note that in both cases, the exponential parameters are well-defined only if the rates and weights are strictly greater than 0 — in practice, however, this is not a significant limitation.

We claim that the joint distribution of a multivariate Poisson mixture model $p(\mathbf{n}, k)$ can be reparameterized in the exponential family form

$$p(\boldsymbol{n}, k) = \frac{e^{\boldsymbol{\theta}_N \cdot \boldsymbol{n} + \boldsymbol{\theta}_K \cdot \boldsymbol{\delta}(k) + \mathbf{n} \cdot \boldsymbol{\Theta}_{NK} \cdot \boldsymbol{\delta}(k) - \psi_{NK}(\boldsymbol{\theta}_N, \boldsymbol{\theta}_K, \boldsymbol{\Theta}_{NK})}}{\prod_i n_i!}, \tag{8}$$

where $\psi_{NK}(\boldsymbol{\theta}_N, \boldsymbol{\theta}_K, \boldsymbol{\Theta}_{NK}) = \log \sum_k e^{\boldsymbol{\theta}_k \cdot \boldsymbol{\delta}(k) + \psi_N(\boldsymbol{\theta}_N + \boldsymbol{\Theta}_{NK} \cdot \boldsymbol{\delta}(k))}$ is the log-partition function of $p(\mathbf{n}, k)$. To show this, we show how to express the natural parameters $\boldsymbol{\theta}_N, \boldsymbol{\theta}_K$, and $\boldsymbol{\Theta}_{NK}$ as (invertible) functions of the component rate vectors $\boldsymbol{\lambda}_1, \ldots, \boldsymbol{\lambda}_{d_K}$, and the weights $w_1, \ldots, w_{d_K}$. In particular, we set

$$\boldsymbol{\theta}_N = \log \boldsymbol{\lambda}_1, \tag{9}$$

where $\log$ is applied element-wise. Then, for $1 \leq k < d_K$, we set the $k$ th row $\boldsymbol{\theta}_{NK,k}$ of $\boldsymbol{\Theta}_{NK}$ to

$$\boldsymbol{\theta}_{NK,k} = \log \boldsymbol{\theta}_{k+1} - \log \boldsymbol{\lambda}_1, \tag{10}$$

and the $k$ th element of $\boldsymbol{\theta}_K$ to

$$\boldsymbol{\theta}_{K,k} = \log \frac{w_{k+1}}{w_1} + \psi(\boldsymbol{\theta}_N) - \psi_N(\boldsymbol{\theta}_N + \boldsymbol{\Theta}_{NK} \cdot \boldsymbol{\delta}(k)). \tag{11}$$

This reparameterization may then be checked by substituting *Equations 9, 10, and 11* into *Equation 8* to recover the joint distribution of the mixture model $p(\mathbf{n}, k) = p(\mathbf{n} \mid k)p(k) = w_k p(\mathbf{n}; \boldsymbol{\lambda}_K)$; for a more explicit derivation see *Sokoloski, 2019*.

The equation for $p(\mathbf{n}, k)$ ensures that the index-probabilities are given by

$$
\begin{aligned}
p(k) &= w_k = e^{\boldsymbol{\theta}_K \cdot \boldsymbol{\delta}(k) - \psi_{NK}(\boldsymbol{\theta}_N, \boldsymbol{\theta}_K, \boldsymbol{\Theta}_{NK})} \sum_{\mathbf{n}} \frac{e^{\mathbf{n} \cdot (\boldsymbol{\theta}_N + \boldsymbol{\Theta}_{NK} \cdot \boldsymbol{\delta}(k))}}{\prod_i n_i!} \\
&= e^{\boldsymbol{\theta}_K \cdot \boldsymbol{\delta}(k) - \psi_{NK}(\boldsymbol{\theta}_N, \boldsymbol{\theta}_K, \boldsymbol{\Theta}_{NK}) + \psi_N(\boldsymbol{\theta}_N + \boldsymbol{\Theta}_{NK} \cdot \boldsymbol{\delta}(k))}.
\end{aligned}
\tag{12}
$$

Consequently, the component distributions in exponential family form are given by

$$
p(\mathbf{n} \mid k) = \frac{p(\mathbf{n}, k)}{p(k)} = \frac{e^{\mathbf{n} \cdot (\boldsymbol{\theta}_N + \boldsymbol{\Theta}_{NK} \cdot \boldsymbol{\delta}(k)) - \psi_N(\boldsymbol{\theta}_N + \boldsymbol{\Theta}_{NK} \cdot \boldsymbol{\delta}(k))}}{\prod_i n_i!}.
\tag{13}
$$

Observe that $p(\mathbf{n} \mid k)$ is a multivariate Poisson distribution with parameters $\boldsymbol{\theta}_N + \boldsymbol{\Theta}_{NK} \cdot \boldsymbol{\delta}(k)$, so that for $k>1$, the parameters are the sum of $\boldsymbol{\theta}_N$ and row $k-1$ of $\boldsymbol{\Theta}_{NK}$. Because the exponential family parameters are the logarithms of the firing rates of $\mathbf{n}$, each row of $\boldsymbol{\Theta}_{NK}$ modulates the firing rates of $\mathbf{n}$ multiplicatively. When $\boldsymbol{\theta}_N(x)$ depends on a stimulus and we consider the component distributions $p(\mathbf{n} \mid x, k)$, each row of $\boldsymbol{\Theta}_{NK}$ then scales the tuning curves of the baseline population (i.e. $p(\mathbf{n} \mid x, k)$ for $k = 1$); in the neuroscience literature, such scaling factors are typically referred to as gain modulations.

The exponential family form has many advantages. However, it has a less intuitive relationship with the statistics of the model such as the mean and covariance. The most straightforward method to compute these statistics given a model in exponential family form is to first reparameterize it in terms of the weights and component rates, and then evaluate *Equations 5, 6, and 7*.

## CoM-Poisson distributions and their mixtures

Conway-Maxwell (CoM) Poisson distributions decouple the location and shape of count distributions (*Shmueli et al., 2005*; *Stevenson, 2016*; *Chanialidis et al., 2018*). A CoM Poisson model has the form $p(\mathbf{n}; \lambda, \nu) \propto \left(\frac{\lambda^n}{n!}\right)^\nu$. The floor function $\lfloor \lambda \rfloor$ of the location parameter $\lambda$ is the mode of the given distribution. With regards to the shape parameter $\nu$, $p(n; \lambda, \nu)$ is a Poisson distribution with rate $\lambda$ when $\nu = 1$, and is under- or over-dispersed when $\nu>1$ or $\nu<1$, respectively. A CoM-Poisson model $p(n; \lambda, \nu)$ is also an exponential family, with natural parameters $\boldsymbol{\theta}_C = (\nu \log \lambda, -\nu)$, sufficient statistic $\mathbf{s}_C(n) = (n, \log n!)$, and base measure $b(n) = 1$. The log-partition function does not have a closed-form expression, but it can be effectively approximated by truncating the series $\sum_{n=0}^{\infty} e^{\mathbf{s}_C(n) \cdot \boldsymbol{\theta}_C}$ (*Shmueli et al., 2005*). More generally, when we consider a product of independent CoM-Poisson distributions, we denote its log-partition function by $\psi_C(\boldsymbol{\theta}_N, \boldsymbol{\theta}_N^*) = \sum_{i=1}^{d_N} \log \sum_{n=0}^{\infty} e^{n\theta_{N,i} + \log(n)! \theta_{N,i}^*}$, where $\boldsymbol{\theta}_{C,i} = (\theta_{N,i}, \theta_{N,i}^*)$ are the parameters of the $i$ th CoM-Poisson distribution. In this case we can also approximate the log-partition function $\psi_C$ by truncating the $d_N$ constituent series $\sum_{n=0}^{\infty} e^{n\theta_{N,i} + \log(n)! \theta_{N,i}^*}$ in parallel.

We define a multivariate CoM-based (CB) mixture as

$$
p(\mathbf{n}, k) = e^{\boldsymbol{\theta}_N \cdot \mathbf{n} + \boldsymbol{\theta}_N^* \cdot \mathbf{lf}(\mathbf{n}) + \boldsymbol{\theta}_K \cdot \boldsymbol{\delta}(k) + \mathbf{n} \cdot \boldsymbol{\Theta}_{NK} \cdot \boldsymbol{\delta}(k) - \psi_{CK}(\boldsymbol{\theta}_N, \boldsymbol{\theta}_N^*, \boldsymbol{\theta}_K, \boldsymbol{\Theta}_{NK})},
\tag{14}
$$

where $\mathbf{lf}(\mathbf{n}) = (\log(n_1!), \ldots, \log(n_{d_N}!))$ is the vector of log-factorials of the individual spike-counts, and $\psi_{CK}(\boldsymbol{\theta}_N, \boldsymbol{\theta}_N^*, \boldsymbol{\theta}_K, \boldsymbol{\Theta}_{NK}) = \log \sum_k e^{\boldsymbol{\theta}_k \cdot \boldsymbol{\delta}(k) + \psi_C(\boldsymbol{\theta}_N + \boldsymbol{\Theta}_{NK} \cdot \boldsymbol{\delta}(k), \boldsymbol{\theta}_N^*)}$ is the log-partition function. This form ensures that the index-probabilities satisfy

$$
p(k) = e^{\boldsymbol{\theta}_K \cdot \boldsymbol{\delta}(k) - \psi_{CK}(\boldsymbol{\theta}_N, \boldsymbol{\theta}_N^*, \boldsymbol{\theta}_K, \boldsymbol{\Theta}_{NK}) + \psi_C(\boldsymbol{\theta}_N + \boldsymbol{\Theta}_{NK} \cdot \boldsymbol{\delta}(k), \boldsymbol{\theta}_N^*)},
\tag{15}
$$

and consequently that each component distribution $p(\mathbf{n} \mid k)$ is a product of independent CoM Poisson distributions given by

$$
p(\mathbf{n} \mid k) = e^{\mathbf{n} \cdot (\boldsymbol{\theta}_N + \boldsymbol{\Theta}_{NK} \cdot \boldsymbol{\delta}(k)) + \boldsymbol{\theta}_N^* \cdot \mathbf{lf}(\mathbf{n}) - \psi_C(\boldsymbol{\theta}_N + \boldsymbol{\Theta}_{NK} \cdot \boldsymbol{\delta}(k), \boldsymbol{\theta}_N^*)}.
\tag{16}
$$

Observe that, whereas the parameters $\boldsymbol{\theta}_N + \boldsymbol{\Theta}_{NK} \cdot \boldsymbol{\delta}(k)$ of $p(\mathbf{n} \mid k)$ depend on the index $k$, the parameters $\boldsymbol{\theta}_N^*$ of $p(\mathbf{n} \mid k)$ are independent of the index and act exclusively as biases. Therefore, realizing different indices $k$ has the effect increasing or decreasing the location parameters, and thus the

modes of the corresponding CoM-Poisson distributions. As such, although the different components of a CB mixture are not simply rescaled versions of the first component $p(\mathbf{n} \mid k = 1)$, in practice they behave approximately in this manner.

The moments of a CoM-Poisson distribution are not available in closed-form, yet they can also be effectively approximated through truncation. We begin by computing approximate means $\mu_{ik}$ and variances $\sigma_{ik}^2$ of $p(n_i \mid k)$ through truncation, and then the mean of $n_i$ is $\mu_i = \sum_{k=1}^{d_K} p(k)\mu_{ik}$, and its variance is

$$\sigma_i^2 = \bar{\sigma}_i^2 + \sum_{k=1}^{d_K} p(k)(\mu_{ik} - \mu_i)^2, \tag{17}$$

where $\bar{\sigma}_i^2 = \sum_{k=1}^{d_K} p(k)\sigma_{ik}^2$. Similarly to *Equation 7*, the covariance $\sigma_{ij}$ between $n_i$ and $n_j$ is $\sigma_{ij} = \sum_{k=1}^{d_K} p(k)(\mu_{ik} - \mu_i)(\mu_{jk} - \mu_j)$.

By comparing *Equations 6 and 17*, we see that the CB mixture may address the limitations on the variances $\sigma_i^2$ of the IP mixture by setting the average variance $\bar{\sigma}_i^2$ of the components in *Equation 17* to be small, while holding the value of the means $\mu_i$ fixed, and ensuring that the means of the components $\mu_{ik}$ cover a wide range of values to achieve the desired values of $\sigma_i^2$ and $\sigma_{ij}$. Solving the parameters of a CB mixture for a desired covariance matrix is unfortunately not possible since we lack closed-form expressions for the means and variances. Nevertheless, we may justify the effectiveness of the CB strategy by considering the approximations of the components means and variances $\mu_{ik} \approx \lambda_{ik} + \frac{1}{2\nu_{ik}} - \frac{1}{2}$ and $\sigma_{ik}^2 \approx \frac{\lambda_{ik}}{\nu_{ik}}$, which hold when neither $\lambda_{ik}$ or $\nu_{ik}$ are too small (*Chanialidis et al., 2018*). Based on these approximations, observe that when $\nu_{ik}$ is large, $\sigma_{ik}^2$ is small, whereas $\mu_{ik}$ is more or less unaffected. Therefore, in the regime where these approximations hold, a small value for $\bar{\sigma}_i^2$ can be achieved by reducing the parameters $\nu_{ik}$, without significantly restricting the values of $\mu_{ik}$ or $\mu_i$.

## Fisher information of a minimal CM

The Fisher information (FI) of an encoding model $p(\mathbf{n} \mid x)$ with respect to $x$ is $I(x) = \sum_{\mathbf{n}} p(\mathbf{n} \mid x)(\partial_x \log p(\mathbf{n} \mid x))^2$ (*Cover and Thomas, 2006*). With regard to the FI of a minimal CM,

$$\partial_x \log p(\mathbf{n} \mid x) = \frac{\sum_k \partial_x p(\mathbf{n}, k \mid x)}{p(\mathbf{n} \mid x)} = \frac{\sum_k \partial_x e^{\boldsymbol{\theta}_N(x) \cdot \mathbf{n} + \boldsymbol{\theta}_N^* \cdot \mathbf{lf}(\mathbf{n}) + \boldsymbol{\theta}_K \cdot \boldsymbol{\delta}(k) + \mathbf{n} \cdot \boldsymbol{\Theta}_{NK} \cdot \boldsymbol{\delta}(k) - \psi_{CK}(\boldsymbol{\theta}_N(x), \boldsymbol{\theta}_N^*, \boldsymbol{\theta}_K, \boldsymbol{\Theta}_{NK})}}{p(\mathbf{n} \mid x)}$$
$$= \partial_x(\boldsymbol{\theta}_N(x) \cdot \mathbf{n} - \psi_{CK}(\boldsymbol{\theta}_N(x), \boldsymbol{\theta}_N^*, \boldsymbol{\theta}_K, \boldsymbol{\Theta}_{NK})) \frac{\sum_k p(\mathbf{n}, k \mid x)}{p(\mathbf{n} \mid x)} = \partial_x \boldsymbol{\theta}_N(x) \cdot (\mathbf{n} - \boldsymbol{\mu}_N(x)),$$

where $\partial_x \psi_{CK}(\boldsymbol{\theta}_N(x), \boldsymbol{\theta}_N^*, \boldsymbol{\theta}_K, \boldsymbol{\Theta}_{NK}) = \boldsymbol{\mu}_N(x) \cdot \partial_x \boldsymbol{\theta}_N(x)$ follows from the chain rule and properties of the log-partition function (*Wainwright and Jordan, 2008*). Therefore

$$I(x) = \sum_{\mathbf{n}} p(\mathbf{n} \mid x)(\partial_x \boldsymbol{\theta}_N(x) \cdot (\mathbf{n} - \boldsymbol{\mu}_N(x)))^2 = \partial_x \boldsymbol{\theta}_N(x) \cdot \boldsymbol{\Sigma}_N(x) \cdot \partial_x \boldsymbol{\theta}_N(x),$$

where $\boldsymbol{\Sigma}_N(x)$ is the covariance matrix of $p(\mathbf{n} \mid x)$. Moreover, because $\partial_x \boldsymbol{\theta}_N(x) = \boldsymbol{\Sigma}_N^{-1}(x) \cdot \partial_x \boldsymbol{\mu}(x)$ (*Wainwright and Jordan, 2008*), the FI of a minimal CM may also be expressed as $I(x) = \partial_x \boldsymbol{\mu}_N(x) \cdot \boldsymbol{\Sigma}_N^{-1}(x) \cdot \partial_x \boldsymbol{\mu}_N(x)$, which is the linear Fisher information (*Beck et al., 2011b*).

Note that when calculating the FI or other quantities based on the covariance matrix, IP-CMs have the advantage that their covariance matrices tend to have large diagonal elements and are thus inherently well-conditioned. Because decoding performance is not significantly different between IP- and CB-CMs (see *Table 3*), IP-CMs may be preferable when well-conditioned covariance matrices are critical. Nevertheless, the covariance matrices of CB mixtures can be made well-conditioned by applying standard techniques.

## Expectation-maximization for CMs

Expectation-maximization (EM) is an algorithm that maximizes the likelihood of a latent variable model given data by iterating two steps: generating model-based expectations of the latent variables, and maximizing the complete log-likelihood of the model given the data and latent

expectations. Although the maximization step optimizes the *complete* log-likelihood, each iteration of EM is guaranteed to not decrease the *data* log-likelihood as well (**Neal and Hinton, 1998**).

EM is arguably the most widely applied algorithm for fitting finite mixture models (**McLachlan et al., 2019**). As a form of latent variable exponential family, the expectation step for a finite mixture model reduces to computing average sufficient statistics, and the maximization step is a convex optimization problem (**Wainwright and Jordan, 2008**). In general, the average sufficient statistics, or mean parameters, correspond to (are dual to) the natural parameters of an exponential family, and where we denote natural parameters with θ, we denote their corresponding mean parameters with η.

Suppose we are given a dataset $(\mathbf{n}^{(1)}, \ldots, \mathbf{n}^{(d_T)})$ of neural spike-counts, and a CB mixture with natural parameters $\boldsymbol{\theta}_N$, $\boldsymbol{\theta}_N^*$, $\boldsymbol{\theta}_K$, and $\boldsymbol{\Theta}_{NK}$ (see **Equation 14**). The expectation step for this model reduces to computing the data-dependent mean parameters $\boldsymbol{\eta}_K^{(i)}$ given by

$$\boldsymbol{\theta}_K^{(i)} = \boldsymbol{\theta}_K + \mathbf{n}^{(i)} \cdot \boldsymbol{\Theta}_{NK}, \quad \eta_{K,k}^{(i)} = \frac{e^{\theta_{K,k}^{(i)}}}{1 + \sum_l e^{\theta_{K,l}^{(i)}}},$$

for all $0 < i \le d_T$. The mean parameters $\boldsymbol{\eta}_K^{(i)}$ are the averages of the sufficient statistic $\delta_k(k)$ under the distribution $p(k \mid \mathbf{n}^{(i)})$, and are what we use to complete the log-likelihood since we do not observe $k$.

Given $\boldsymbol{\eta}_K^{(i)}$, the maximization step of a CB mixture thus reduces to maximizing the complete log-likelihood $\sum_{i=1}^{d_T} \mathcal{L}(\boldsymbol{\theta}_K, \boldsymbol{\theta}_N, \boldsymbol{\theta}_N^*, \boldsymbol{\Theta}_{NK}, \boldsymbol{\eta}_K^{(i)}, \mathbf{n}^{(i)})$, where we substitute $\boldsymbol{\eta}_K^{(i)}$ into the place of $\boldsymbol{\delta}(k)$ in **Equation 14**, such that

$$\mathcal{L}(\boldsymbol{\theta}_K, \boldsymbol{\theta}_N, \boldsymbol{\theta}_N^*, \boldsymbol{\Theta}_{NK}, \boldsymbol{\eta}_K^{(i)}, \mathbf{n}^{(i)}) = \boldsymbol{\theta}_N \cdot \mathbf{n}^{(i)} + \boldsymbol{\theta}_N^* \cdot \mathbf{lf}(\mathbf{n}^{(i)}) + \boldsymbol{\theta}_K \cdot \boldsymbol{\eta}_K^{(i)} + \mathbf{n}^{(i)} \cdot \boldsymbol{\Theta}_{NK} \cdot \boldsymbol{\eta}_K^{(i)} - \psi_{CK}(\boldsymbol{\theta}_N, \boldsymbol{\theta}_N^*, \boldsymbol{\theta}_K, \boldsymbol{\Theta}_{NK}).$$

This objective may be maximized in closed-form for an IP mixture (**Karlis and Meligkotsidou, 2007**), but this is not the case when the model has CoM-Poisson shape parameters or depends on the stimulus. Nevertheless, solving the resulting maximization step is still a convex optimization problem (**Wainwright and Jordan, 2008**), and may be approximately solved with gradient ascent. Doing so requires that we first compute the mean parameters $\boldsymbol{\eta}_N$, $\boldsymbol{\eta}_N^*$, $\boldsymbol{\eta}_K$, and $\mathbf{H}_{NK}$ that are dual to $\boldsymbol{\theta}_N$, $\boldsymbol{\theta}_N^*$, $\boldsymbol{\theta}_K$, and $\boldsymbol{\Theta}_{NK}$, respectively.

We compute the mean parameters by evaluating

$$\theta_{K,k}^{\dagger} = \theta_{K,k} + \psi_C(\boldsymbol{\theta}_N + \boldsymbol{\Theta}_{NK} \cdot \boldsymbol{\delta}(k), \boldsymbol{\theta}_N^*) - \psi(\boldsymbol{\theta}_N) \quad \eta_{K,k} = \frac{e^{\theta_{K,k}^{\dagger}}}{1 + \sum_{k=1}^{d_K-1} e^{\theta_{K,k}^{\dagger}}}, \quad \mu_{jk} = \sum_{n_j=0}^{\infty} n_j \, p(n_j \mid k),$$

$$\eta_{N,j}^* = \sum_{k=1}^{d_K} p(k) \sum_{n_j=0}^{\infty} \log n_j! \, p(n_j \mid k), \qquad \eta_{N,j} = \sum_{k=1}^{d_K} p(k) \mu_{jk}, \qquad \eta_{NK,jk} = \eta_{K,k} \mu_{j(k+1)},$$

where $\eta_{K,k}$ is the $k$ th element of $\boldsymbol{\eta}_K$, $\eta_{N,j}$ is the $j$ th element of $\boldsymbol{\eta}_N$, $\eta_{N,j}^*$ is the $j$ th element of $\boldsymbol{\eta}_N^*$, and $\eta_{NK,jk}$ is the $j$ th element of the $k$ th column of $\mathbf{H}_{NK}$. Note as well that we truncate the series $\sum_{n_j} n_j \, p(n_j \mid k)$ and $\sum_{n_j} \log n_j! \, p(n_j \mid k)$ to approximate $\mu_{jk}$ and $\eta_{N,j}^*$. Given these mean parameters, we may then express the gradients of $\mathcal{L}^{(i)} = \mathcal{L}(\boldsymbol{\theta}_K, \boldsymbol{\theta}_N, \boldsymbol{\theta}_N^*, \boldsymbol{\Theta}_{NK}, \boldsymbol{\eta}_{K,i}, \mathbf{n}^{(i)})$ as

$$\partial_{\theta_N} \mathcal{L}^{(i)} = \mathbf{n}^{(i)} - \boldsymbol{\eta}_N, \qquad \partial_{\theta_N^*} \mathcal{L}^{(i)} = \mathbf{lf}(\mathbf{n}^{(i)}) - \boldsymbol{\eta}_N^*,$$
$$\partial_{\theta_K} \mathcal{L}^{(i)} = \boldsymbol{\eta}_K^{(i)} - \boldsymbol{\eta}_K, \quad \partial_{\Theta_{NK}} \mathcal{L}^{(i)} = \mathbf{n}^{(i)} \otimes \boldsymbol{\eta}_K^{(i)} - \mathbf{H}_{NK},$$

where $\otimes$ is the outer product operator, and where the second term in each equation follows from the fact that the derivative of $\psi_{CK}$ with respect to $\boldsymbol{\theta}_N$, $\boldsymbol{\theta}_N^*$, $\boldsymbol{\theta}_K$, or $\boldsymbol{\Theta}_{NK}$ yields the dual parameters $\boldsymbol{\eta}_N$, $\boldsymbol{\eta}_N^*$, $\boldsymbol{\eta}_K$, and $\mathbf{H}_{NK}$, respectively. By ascending the gradients of $\sum_{i=1}^{d_T} \mathcal{L}^{(i)}$ until convergence, we approximate a single iteration of the EM algorithm for a CB mixture.

Finally, if our dataset $((\mathbf{n}^{(1)}, x^{(1)}), \ldots, (\mathbf{n}^{(d_T)}, x^{(d_T)}))$ includes stimuli $x$, and the parameters $\boldsymbol{\theta}_N$ depend on the stimulus, then the gradients of the parameters of $\theta_N$ must also be computed. For a von Mises CM where $\boldsymbol{\theta}_N(x) = \boldsymbol{\theta}_N^0 + \boldsymbol{\Theta}_{NX} \cdot \mathbf{vm}(x)$, the gradients are given by

$$\partial_{\boldsymbol{\theta}_N^0} \mathcal{L}^{(i)} = \partial_{\boldsymbol{\theta}_N^{(i)}} \mathcal{L}^{(i)}, \quad \partial_{\boldsymbol{\Theta}_{NX}} \mathcal{L}^{(i)} = \partial_{\boldsymbol{\theta}_N^{(i)}} \mathcal{L}^{(i)} \otimes \mathbf{vm}(x^{(i)}),$$

where $\boldsymbol{\theta}_N^{(i)} = \boldsymbol{\theta}_N(x^{(i)})$ is the output of $\boldsymbol{\theta}_N$ at $x^{(i)}$. Although in this paper we restrict our applications

to Von Mises or discrete tuning curves for one-dimensional stimuli, this formalism can be readily extended to the case where the baseline parameters $\boldsymbol{\theta}_N(x)$ are a generic nonlinear function of the stimulus, represented by a deep neural network. Then, the gradients of the parameters of $\boldsymbol{\theta}_N$ can be computed through backpropagation, and $\partial_{\boldsymbol{\theta}_N^{(i)}}\mathcal{L}^{(i)}$ is the error that must be backpropagated through the network to compute the gradients.

If we ignore stimulus dependence, the single most computationally intensive operation in each gradient ascent step is the computation of the outer product when evaluating $\partial_{\boldsymbol{\Theta}_{NK}}\mathcal{L}^{(i)}$, which has a time complexity of $\mathcal{O}(d_K d_N)$. As such, the training algorithm scales linearly in the number of neurons, and CMs could realistically be applied to populations of tens to hundreds of thousands of neurons. That being said, larger values of $d_K$ will typically be required to maximize performance in larger populations, and fitting the model to larger populations typically requires larger datasets and more EM iterations.

## CM initialization and training procedures

To fit a CM to a dataset $((\mathbf{n}^{(1)}, x^{(1)}), \ldots, (\mathbf{n}^{(d_T)}, x^{(d_T)}))$, we first initialize the CM and then optimize its parameters with our previously described EM algorithm. Naturally, initialization depends on exactly which form of CM we consider, but in general we first initialize the baseline parameters $\boldsymbol{\theta}_N$, then add the categorical parameters $\boldsymbol{\theta}_K$ and mixture component parameters $\boldsymbol{\Theta}_{NK}$. When training CB-CMs we always first train an IP-CM, and so the initialization procedure remains the same for IP and CB models.

To initialize a von Mises CM with $d_N$ neurons, we first fit $d_N$ independent, von Mises-tuned neurons by maximizing the log-likelihood $\sum_{i=1}^{d_T} \log p(\mathbf{n}^{(i)} \mid x^{(i)})$ of $\boldsymbol{\theta}_N(x) = \boldsymbol{\theta}_N^0 + \boldsymbol{\Theta}_{NX} \cdot \mathbf{vm}(x)$. This is a convex optimization problem and so can be easily solved by gradient ascent, in particular by following the gradients

$$\partial_{\boldsymbol{\theta}_N^0} \sum_{i=1}^{d_T} \log p(\mathbf{n}^{(i)} \mid x^{(i)}) = \sum_{i=1}^{d_T} \mathbf{n}^{(i)} - \log(\boldsymbol{\theta}_N(x^{(i)})),$$

$$\partial_{\boldsymbol{\Theta}_{NX}} \sum_{i=1}^{d_T} \log p(\mathbf{n}^{(i)} \mid x^{(i)}) = \sum_{i=1}^{d_T} \log(\mathbf{n}^{(i)} - \log \boldsymbol{\theta}_N(x^{(i)})) \otimes \mathbf{vm}(x^{(i)}),$$

to convergence. For both discrete and maximal CMs, where there are $d_X$ distinct stimuli, we initialize $\boldsymbol{\theta}_N(x) = \boldsymbol{\theta}_N^0 + \boldsymbol{\Theta}_{NX} \cdot \boldsymbol{\delta}(x)$ by computing the average rate vector at each stimulus-condition and creating a lookup table for these rate vectors. Formally, where $x_l$ is the $l$ th stimulus value for $0 < l \leq d_X$, we may express the $l$ th rate vector as $\boldsymbol{\lambda}_l = \frac{1}{\sum_{i=1}^{d_T} \delta(x_l, x^{(i)})} \sum_{i=1}^{d_T} \delta(x_l, x^{(i)}) \mathbf{n}^{(i)}$, where $\delta(x_l, x^{(i)})$ is one when $x_l = x^{(i)}$, and 0 otherwise. We then construct a lookup table for these rate vectors in exponential family form by setting $\boldsymbol{\theta}_N^0 = \log \boldsymbol{\lambda}_1$, and by setting the $l$ th row $\boldsymbol{\theta}_{NX,l}$ of $\boldsymbol{\Theta}_{NX}$ to $\boldsymbol{\theta}_{NX,l} = \log \boldsymbol{\lambda}_{l+1} - \log \boldsymbol{\lambda}_1$.

In general, we initialize the parameters $\boldsymbol{\theta}_K$ by sampling the weights $w_1, \ldots, w_{d_K}$ of a categorical distribution from a Dirichlet distribution with a constant concentration of 2, and converting the weights into the natural parameters of a categorical distribution $\boldsymbol{\theta}_K$. For discrete and maximal CMs, we initialize the modulations $\boldsymbol{\Theta}_{NK}$ by generating each element of $\boldsymbol{\Theta}_{NK}$ from a uniform distribution over the range $[-0.0001, 0.0001]$. For von Mises CMs we initialize each row $\boldsymbol{\theta}_{NK,k}$ of $\boldsymbol{\Theta}_{NK}$ as shifted sinusoidal functions of the preferred stimuli of the independent von Mises neurons. That is, given $\boldsymbol{\theta}_N^0$ and $\boldsymbol{\Theta}_{NX}$, we compute the preferred stimulus of the $i$ th neuron given by $\rho_i = \mathrm{atan2}(\boldsymbol{\theta}_N^0 + \boldsymbol{\theta}_{NX,i})$, where $\boldsymbol{\theta}_{NX,i}$ is the $i$ th row of $\boldsymbol{\Theta}_{NX}$. We then set the $i$ th element $\theta_{NK,k,i}$ of $\boldsymbol{\theta}_{NK,k}$ to $\theta_{NK,k,i} = 0.2 \sin(\rho_i + \frac{k}{360}^\circ)$. Initializing von Mises CMs in this way ensures that each modulation has a unique peak as a function of preferred stimuli, which helps differentiate the modulations from each other, and in our experience improves training speed.

With regard to training, the expectation step in our EM algorithm may be computed directly, and so the only challenge is solving the maximization step. Although the optimal solution strategy depends on the details of the model and data in question, in the context of this paper we settled on a strategy that is sufficient for all simulations we perform. For each model we perform a total of $d_I = 500$ EM iterations, and for each maximization step we take $d_S = 100$ gradient ascent steps with the Adam gradient ascent algorithm (*Kingma and Ba, 2014*) with the default momentum parameters (see *Kingma and Ba, 2014*). We restart the Adam algorithm at each iteration of EM and gradually

reduce the learning rate. Where $\epsilon^+ = 0.002$ and $\epsilon^- = 0.0005$ are the initial and final learning rates, we set the learning rate $\epsilon_t$ at EM iteration $t$ to

$$\epsilon_t = \exp\left(\frac{(d_I - 1 - t)\log(\epsilon^+) + t\log(\epsilon^-)}{d_I - 1}\right),$$

where we assume $t$ starts at 0 and ends at $d_I - 1$.

Because we must evaluate large numbers of truncated series when working with CB-CMs, training times are typically one to two orders of magnitude greater. To minimize training time of CB-CMs over the $d_I$ EM iterations, we therefore first train a IP-CM for $0.8d_I$ iterations. We then equate the parameters $\boldsymbol{\theta}_N$, $\boldsymbol{\theta}_K$, and $\boldsymbol{\Theta}_{NK}$ of the IP-CM (see *Equation 8*) with a CB-CM (see *Equation 14*) and set $\boldsymbol{\theta}_N^* = -\mathbf{1}$, which ensures that resulting CB model has the same density function $p(\mathbf{n}, k \mid x)$ as the original IP model. We then train the CB-CM for $0.2d_I$ iterations. We found this strategy results in practically no performance loss, while greatly reducing training time.

## Strategies for choosing the CM form and latent structure

There are a few choices with regards to the form of the model than one must make when applying a CM: The form of the dependence, whether or not to use the CoM-based (CB) extension, and the number of components $d_K$. The form of the dependence is very open-ended, yet should be fairly clear from the problem context: one should use a minimal model if one wishes to make use of its mathematical features, and otherwise a maximal model may provide better performance. If one wishes to interpolate between stimulus conditions, or the number of stimulus-conditions in the data is high, then a continuous stimulus-dependence model (e.g. von Mises tuning curves) should be used, otherwise discrete tuning curves may provide better performance. Finally, if one wishes to model correlations in a complex neural circuit, one may use for example a deep neural network, and induce correlations in the output layer with the theory of CMs.

Similarly, CB-CMs have clear advantages for modelling individual variability, and as we show in Appendix 2, this includes higher-order variability. Nevertheless, from the perspective of decoding performance, IP-CMs and CB-CMs perform more-or-less equally well, and training CB-CMs is more computationally intensive. As such, IP-CMs may often be the better choice.

The number of components $d_K$ provides a fine-grained method of adjusting model performance. If the goal is to maximize predictive encoding performance, then the standard way to do this is to choose a $d_K$ that maximizes the cross-validated log-likelihood, as we demonstrated in *Figure 5*. Nevertheless, one may rather aim to maximize decoding performance, in which case maximizing the cross-validated log-posterior may be a more appropriate objective. In both cases, for very large populations of neurons, choosing a $d_K$ solely to maximize performance may be prohibitively, computationally expensive. As demonstrated in *Figure 5* and Appendix 3, a small $d_K$ can achieve a large fraction of the performance gain of the optimal $d_K$, and choosing a modest $d_K$ that achieves qualitatively acceptable performance may prove to be the most productive strategy.

## CM parameter selection for simulations

In the section Extended Poisson mixture models capture stimulus-dependent response statistics and the section Conditional mixtures facilitate accurate and efficient decoding of neural responses we considered minimal CB-CMs with randomized parameters $\boldsymbol{\theta}_N(x)$, $\boldsymbol{\theta}_N^*$, $\boldsymbol{\theta}_K$, and $\boldsymbol{\Theta}_{NK}$, which for simplicity we refer to as models 1 and 2, respectively. We construct randomized CMs piece by piece, in a similar fashion to our initialization procedure.

Firstly, where $d_N$ is the number of neurons, we tile their preferred stimuli $\rho_i$ over the circle such that $\rho_i = \frac{i}{d_N}360°$. We then generate the concentration $\kappa_i$ and gain $\gamma_i$ of the $i$ th neuron by sampling from normal distributions in log-space, such that $\log\kappa_i \sim \mathrm{N}(-0.1, 0.2)$, and $\log\gamma_i \sim \mathrm{N}(0.2, 0.1)$. Finally, for von Mises baseline parameters $\boldsymbol{\theta}_N(x) = \boldsymbol{\theta}_N^0 + \boldsymbol{\Theta}_{NX} \cdot \mathbf{vm}(x)$, we set each row $\boldsymbol{\theta}_{NX,i}$ of $\boldsymbol{\Theta}_{NX}$ to $\boldsymbol{\theta}_{NX,i} = (\kappa_i \cos\rho_i, \kappa_i \sin\rho_i)$, and each element $\theta_{N,i}^0$ of $\boldsymbol{\theta}_N^0$ to $\theta_{N,i}^0 = \log\gamma_i - \psi_X(\boldsymbol{\theta}_{NX,i})$, where $\psi_X$ is the logarithm of the modified Bessel function of order 0, which is the log-partition function of the von Mises distribution.

We then set $\boldsymbol{\theta}_K = \mathbf{0}$, and generated each element $\theta_{NK,i,k}$ of the modulation matrix $\boldsymbol{\theta}_{NK}$ in the same matter as the gains, such that $\theta_{NK,i,k} \sim \mathrm{N}(0.2, 0.1)$. Finally, to generate random CB parameters we generate each element $\theta_{N,i}^*$ of $\boldsymbol{\theta}_N^*$ from a uniform distribution, such that $\theta_{N,i}^* \sim \mathrm{U}(-1.5, -0.8)$.

Model two entails two more steps. Firstly, when sampling from larger populations of neurons, single modulations often dominate the model activity around certain stimulus values. To suppress this we consider the natural parameters $\boldsymbol{\theta}_K^0(x)$ of $p(k \mid x)$ (see *Equation 15*), and compute the maximum value of these natural parameters over the range of stimuli $\theta_{K,k}^+ = \max_x\{\theta_{K,k}^0(x)\}$. We then set each element $\theta_{K,k}$ of the parameters $\boldsymbol{\theta}_K$ of the CM to $\theta_{K,k} = \bar{\theta}_K^+ - \theta_{K,k}^+$, where $\bar{\theta}_K^+ = \sum_{i=1}^{d_K} \frac{\theta_{K,k}}{d_K}$, which helps ensure that multiple modulations are active at any given $x$. Finally, since model two is a discrete CM, we replace the von Mises baseline parameters with discrete baseline parameters, by evaluating $\boldsymbol{\theta}_N^0 + \boldsymbol{\Theta}_{NX} \cdot \mathbf{vm}(x)$ at each of the $d_X$ valid stimulus-conditions, and assemble the resulting collection of natural parameters into a lookup table in the manner we described in our initialization procedures.

## Decoding models

When constructing a Bayesian decoder for discrete stimuli, we first estimate the prior $p(x)$ by computing the relative frequency of stimulus presentations in the training data. For the given encoding model, we then evaluate $p(\mathbf{n} \mid x)$ at each stimulus condition, and then compute the posterior $p(x \mid \mathbf{n}) \propto p(\mathbf{n} \mid x)p(x)$ by brute-force normalization of $p(\mathbf{n} \mid x)p(x)$. When training the encoding model used for our Bayesian encoders, we only trained them to maximize encoding performance as previously described, and not to maximize decoding performance.

We considered two decoding models, namely the linear network and the artificial neural network (ANN) with sigmoid activation functions. In both cases, the input of the network was a neural response vector, and the output the natural parameters $\boldsymbol{\theta}_X$ of a categorical distribution. The form of the linear network was $\boldsymbol{\theta}_X(\mathbf{n}) = \boldsymbol{\theta}_X + \boldsymbol{\Theta}_{XN} \cdot \mathbf{n}$, and is otherwise fully determined by the structure of the data. For the ANN on the other hand, we had to choose both the number of hidden layers, and the number of neurons per hidden layer. We cross-validated the performance of both 1 and 2 hidden layer models, over a range of sizes from 100 to 2000 neurons. We found the performance of the networks with two hidden layers generally exceeded that of those with one hidden layer, and that 700 and 600 hidden neurons was optimal for the awake and anaesthetized networks, respectively.

Given a dataset $((\mathbf{n}^{(1)}, x^{(1)}), \ldots, (\mathbf{n}^{(d_T)}, x^{(d_T)}))$, we optimized the linear network and the ANN by maximizing $\sum_{i=1}^{d_T} \log p(x^{(i)} \mid \mathbf{n}^{(i)})$ via stochastic gradient ascent. We again used the Adam optimizer with default momentum parameters, and used a fixed learning rate of 0.0003, and randomly divided the dataset into minibatches of 500 data points. We also used early stopping, where for each fold of our 10-fold cross-validation simulation, we partitioned the dataset into 80% training data, 10% test data, and 10% validation data, and stopped the simulation when performance on the test data declined from epoch to epoch.

## Experimental design

Throughout this paper, we demonstrate our methods on two sets of parallel response recordings in macaque primary visual cortex (V1). The stimuli were drifting full contrast gratings at nine distinct orientations spread evenly over the half-circle from 0° to 180° (2° diameter, two cycles per degree, 2.5 Hz drift rate). Stimuli were generated with custom software (EXPO by P. Lennie) and displayed on a cathode ray tube monitor (Hewlett Packard p1230; 1024 × 768 pixels, with ∼ 40 cd/m² mean luminance and 100 Hz frame rate) viewed at a distance of 110 cm (for anaesthetized dataset) or 60 cm (for awake dataset). Grating orientations were randomly interleaved, each presented for 70 ms (for anaesthetized dataset) or 150 ms (for awake dataset), separated by a uniform gray screen (blank stimulus) for the same duration. Stimuli were centered in the aggregate spatial receptive field of the recorded units.

Neural activity from the superficial layers of V1 was recorded from a 96 channel microelectrode array (400 μm spacing, 1 mm length). (400 μm spacing, 1 mm length). Standard methods for waveform extraction and pre-processing were applied (see *Aschner et al., 2018*). We computed spike counts in a fixed window with length equal to the stimulus duration, shifted by 50 ms after stimulus onset to account for onset latency. We excluded from further analyses all neurons that were not driven by any stimulus above baseline + 3std.

In the first dataset, the monkey was awake and performed a fixation task. Methods and protocols are as described in *Festa et al., 2020*. There were $d_T = 3,168$ trials of the responses of $d_N = 43$ neurons in the dataset. We refer to this dataset as the awake V1 dataset.

In the second dataset, the monkey was anaesthetized and there were $d_T = 10,800$ trials of the responses of $d_N = 70$ neurons; we refer to this dataset as the anaesthetized V1 dataset. The protocol and general methods employed for the anaesthetized experiment have been described previously (*Smith and Kohn, 2008*).

All procedures were approved by the Institutional Animal Care and Use Committee of the Albert Einstein College of Medicine, and were in compliance with the guidelines set forth in the National Institutes of Health Guide for the Care and Use of Laboratory Animals under protocols 20180308 and 20180309 for the awake and anaesthetized macaque recordings, respectively.

## Code

All code used to run the simulations and generate the figures, as well as the awake and anaesthetized datasets, are available at the Git repository https://gitlab.com/sacha-sokoloski/neural-mixtures (*Sokoloski, 2021*; copy archived at swh:1:rev:8e82799f8934c47961ea02c5b7c25bd952abb961). Instructions are provided for installation, and scripts are provided that may be run on alternative datasets with a similar structure to what we have considered in this manuscript without modifying the code.

## Acknowledgements

We thank all the members of the labs of Ruben Coen-Cagli and Adam Kohn for their regular feedback and support.

## Additional information

### Funding

| Funder | Grant reference number | Author |
| --- | --- | --- |
| National Institutes of Health | EY030578 | Ruben Coen-Cagli |
| National Institutes of Health | EY02826 | Sacha Sokoloski Amir Aschner |
| National Institutes of Health | EY016774 | Amir Aschner |

The funders had no role in study design, data collection and interpretation, or the decision to submit the work for publication.

### Author contributions

Sacha Sokoloski, Conceptualization, Software, Formal analysis, Validation, Investigation, Visualization, Methodology, Writing - original draft, Writing - review and editing; Amir Aschner, Data curation, Writing - review and editing, Data Collection; Ruben Coen-Cagli, Conceptualization, Supervision, Funding acquisition, Methodology, Project administration, Writing - review and editing

### Author ORCIDs

Sacha Sokoloski (ID) https://orcid.org/0000-0003-4166-1772
Ruben Coen-Cagli (ID) https://orcid.org/0000-0003-2052-5894

### Ethics

Animal experimentation: All procedures were approved by the Institutional Animal Care and Use Committee of the Albert Einstein College of Medicine, and were in compliance with the guidelines set forth in the National Institutes of Health Guide for the Care and Use of Laboratory Animals under protocols 20180308 and 20180309 for the awake and anaesthetized macaque recordings, respectively.

### Decision letter and Author response

Decision letter https://doi.org/10.7554/eLife.64615.sa1
Author response https://doi.org/10.7554/eLife.64615.sa2

## Additional files

### Supplementary files

• Transparent reporting form

### Data availability

All data used in this study is available at the Git repository (https://gitlab.com/sacha-sokoloski/neural-mixtures; copy archived at https://archive.softwareheritage.org/swh:1:rev:8e82799f8934c47961ea02c5b7c25bd952abb961). This includes experimental data used for model validation, and code for running analyses and simulations.

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

# Appendix 1

## Comparing conditional mixtures with factor analysis

In Conditional mixtures effectively model neural responses in macaque V1, we assess encoding performance with the cross-validated, average log-likelihood of the given conditional mixture (CM) on the given dataset. However, in some cases, one might only be concerned with how well a model captures particular statistics of a dataset. In particular, response models based on Gaussian distributions treat spike-counts as continuous values, and assign positive probability to both negative and non-integer values. Although their log-likelihood performance consequently tends to suffer relative to spike-count models, they can still prove highly effective at capturing the mean and covariance of data.

Here, we compare CMs with factor analysis (FA), which is widely applied to modelling neural responses (*Santhanam et al., 2009*; *Cowley et al., 2016*; *Semedo et al., 2019*). FAs model data as Gaussian distributed, and have a latent structure that facilitates both interpretability and predictive performance. The easiest way to design an encoding model based on FA is with a simple lookup-table, so that we fit an independent FA model at each stimulus condition. This is also how we define maximal CMs, and so to keep our comparison straightforward we compare FA encoding models with maximal CMs. In particular, we compare FA to both independent Poisson (IP) and CoM-Based (CB) maximal CMs on how well they capture response statistics on the two datasets from the article (anaesthetized and awake macaque V1). In general, we trained the CMs with expectation-maximization (EM) as described in Materials and methods, and the FA model with standard EM.

In *Appendix 1—figure 1A and B*, we depict scatter plots that compare the data noise correlations from our awake and anaesthetized datasets at the stimulus orientation $x = 40°$, to the noise correlations learned by CB and IP mixtures, and FA trained on the complete datasets. Each model was defined with $d_K = 5$ latent states/dimensions. We also state the coefficient of determination $r^2$ for each model, and see that although all models perform comparably on the anaesthetized data, FA has a clear advantage over the mixture models on the awake data. To see if this performance advantage holds on held-out data, in *Appendix 1—figure 1C and D* we depict the results of 10-fold cross-validation of the coefficient of determination $r^2$ between the data noise correlations and the various model noise correlations over all nine stimulus orientations, as a function of the number of latant states/dimensions $d_K$. We see that the predictive performance advantage of FA on the awake data is small, and that CB-CMs exceed the performance of FA on anaesthetized data. At the same time, FA achieves peak performance on both datasets with a smaller number of parameters. Nevertheless, FA is designed precisely to capture second-order correlations, and that our mixture models achieve comparable performance speaks favourably to the overall strengths of the mixture model approach.

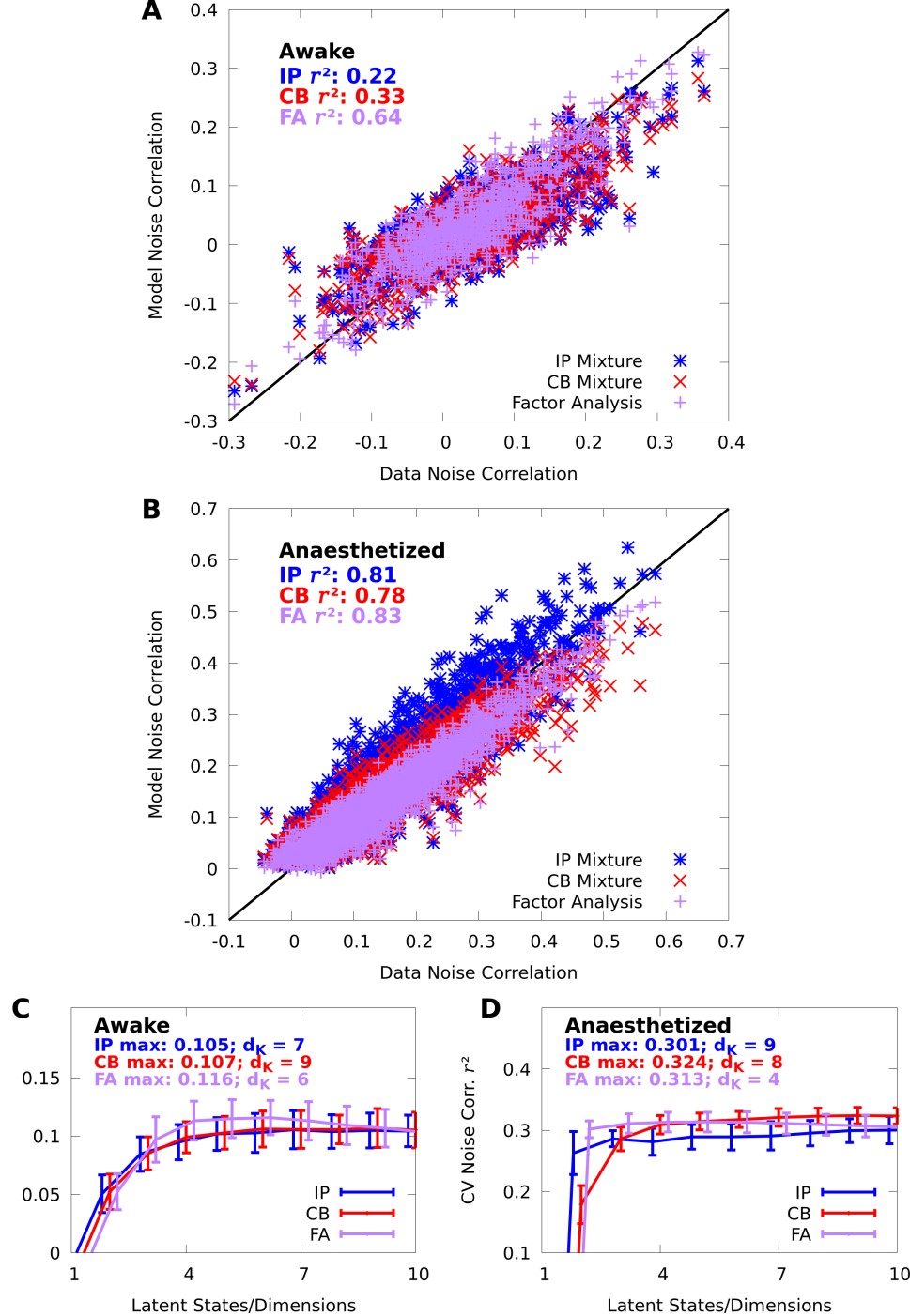

**Appendix 1—figure 1.** Mixture models capture spike-count correlations. (**A**, **B**) Scatter plots of the data noise correlations versus the noise correlations modelled by IP (blue) and CB (red) mixtures, and FA (purple), in both the awake (**A**) and anaesthetized (**B**) datasets at orientation $x = 40°$. Each point represent a pair of neurons ($d_N = 43$ and $d_N = 70$ neurons in the awake and anaesthetized datasets, respectively). (**C**, **D**) We evaluate the noise correlation $r^2$ over all stimulus-orientations with 10-fold cross-validation. We plot the average (lines) and standard error (error bars) of the cross-validated noise correlation $r^2$ as a function of number of latent states/dimensions $d_K$, in both the awake (**C**) and anaesthetized (**D**) datasets. We also indicate the peak performance achieved for each model, and requisite number of latent states/dimensions $d_K$.

In *Appendix 1—figure 2*, we depict scatter plots between the data Fano factors (FFs) and learned FFs of our models at stimulus orientation $x = 40°$, and find that both the CB mixture and FA

almost perfectly capture the data FFs. In *Appendix 1—figure 2C–D*, we see that the CB mixture and FA also achieve good cross-validated $r^2$ scores on FFs. Unsurprisingly, however, the IP mixture struggles to effectively capture FFs.

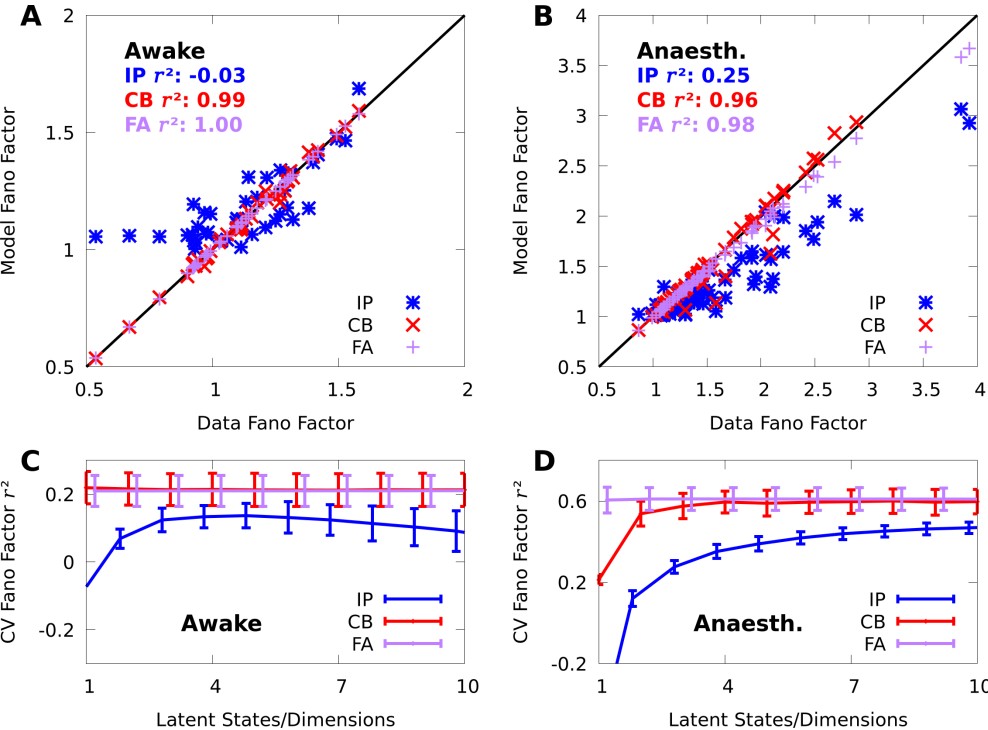

**Appendix 1—figure 2.** Mixture models capture spike-count Fano factors. We repeat the analyses from Appendix References Figure 7 on Fano factors (FFs). (**A**, **B**) Scatter plots of the data FFs versus the FFs modelled by IP (blue) and CB mixtures (red), and FA (purple) in both the awake (**A**) and anaesthetized (**B**) datasets at orientation $x = 40°$. (**C**, **D**) As a function of the number of latent states/ dimensions, we plot the average (lines) and standard error (bars) of the cross-validated $r^2$ between the data and modelled Fano factors over all stimulus orientations, in both the awake (**C**) and anaesthetized (**D**) datasets.

## Appendix 2

### Higher order moments and neural decoding

FA-based encoding models are highly effective at capturing the first- and second-order statistics of neural responses, yet in our simulations we found that Bayesian decoders based on FA encoding models perform poorly when compared to the other decoding models considered in Conditional mixtures facilitate accurate and efficient decoding of neural responses. There we evaluate decoding performance by fitting a candidate model to training data, and computing the mean and standard error of the log-posterior at the true stimulus on held-out data. On the awake data FA scores $-0.246 \pm 0.066$, which is comparable to an optimal linear decoder, yet still significantly worse than a nonlinear decoder, or a Bayesian decoder based on a CM. On the anaesthetized data FA scored $-\infty$, as it would occasionally assign numerically 0 posterior probability to the true stimulus; when we filtered out $-\infty$ values from the average, the FA encoder still only achieved performance of $-2.21 \pm 0.31$.

Normal distributions — and the FA observable distribution by extension — are essentially defined by their first- and second-order statistics, which suggests that there are higher-order statistics that are important for decoding that FA cannot capture. The third and fourth order statistics known as the skewness and excess kurtosis measure the asymmetry and heavy-tailedness of a given distribution, respectively. Normal distributions have a skewness and excess kurtosis of 0. Here, we study how well this normality assumption is reflected in our neural recordings, and how well our mixture models capture these higher order statistics.

In *Figure 1*, we present scatter plots of the empirical skewness and kurtosis of all the neurons in our datasets at orientation $x = 40°$, and the model skewness and kurtosis learned by our mixture models. Exactly quantifying the non-normality of higher order moments in multivariate distributions is a complicated and evolving subject (*Mardia and El-atoum, 1976*; *Cain et al., 2017*), nevertheless in *Figure 1* the empirical skewness and kurtosis of the recorded neurons appear to qualitatively deviate from zero. On the awake data, both the CB and IP mixture achieve high $r^2$ when compared with the data skewness (*Figure 1A*) and kurtosis (*Figure 1B*), although the CB mixtures achieves notably better performance. On the anaesthetized data (*Figure 1C and D*), the CB mixture continues to achieve a high $r^2$, but the IP mixture performs extremely poorly; although the disparity in $r^2$ is not immediately apparent in the scatter plots, this is because some of the model skewness of the IP mixtures are outside the plot bounds.

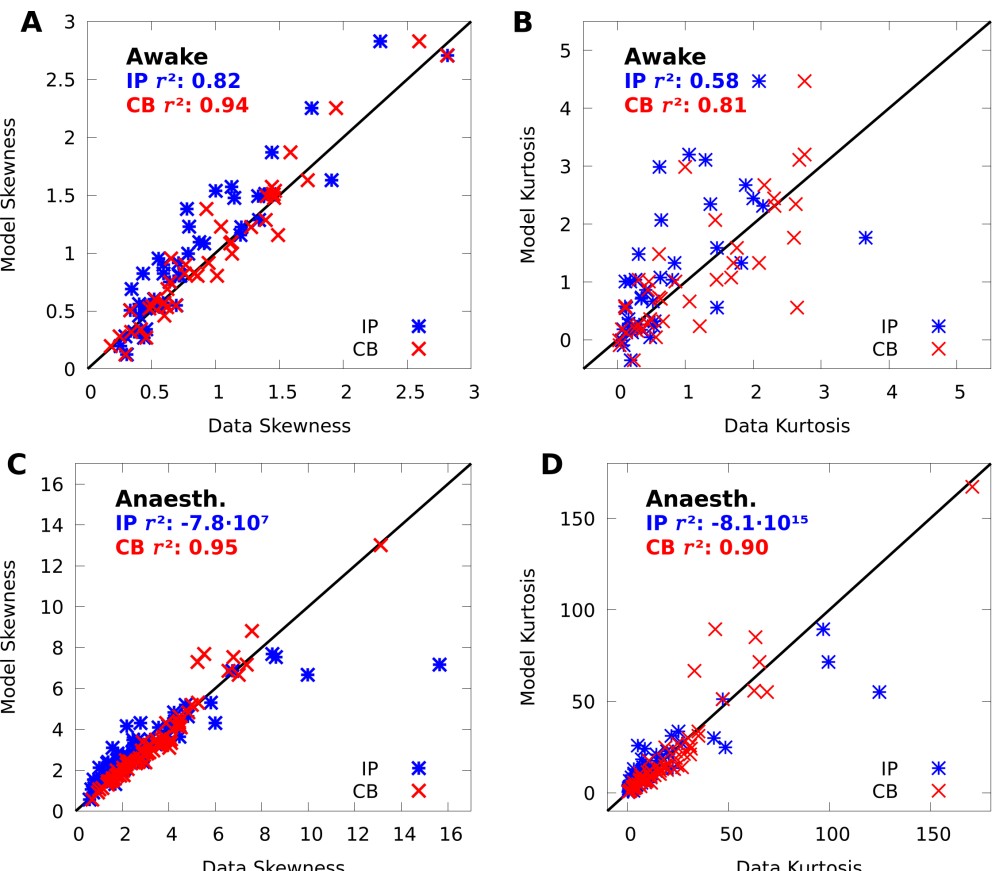

**Appendix 2—figure 1.** CoM-based mixtures capture data skewness and kurtosis. (**A**, **B**) Scatter plots of the data skewness (**A**) and kurtosis (**B**) versus the skewness and kurtosis modelled by IP mixtures (blue) and CB mixtures (red). The skewness and kurtosis of 1 of 43 neurons modelled by the IP mixture were outside the bounds of each scatter plot. C,D: Same as A-B but on the anaesthetized data; the skewness of 11 of 43, and kurtosis of 12 of 43 neurons modelled by the IP mixture were outside the bounds of each scatter plot.

When fit to the complete datasets, and averaged over all stimulus conditions, the CB mixtures achieved a skewness $r^2$ average and standard error of $r^2 = 0.87 \pm 0.08$ and $r^2 = 0.94 \pm 0.03$, and a kurtosis $r^2$ average and standard error of $r^2 = 0.61 \pm 0.20$ and $r^2 = 0.82 \pm 0.13$ on the awake and anaesthetized data, respectively; in contrast, the presence of outliers caused the average scores for the IP mixture to be dramatically negative in all cases. These results suggest that the CoM-based parameters of the CB mixture provide important degrees of freedom for capturing individual variability. That being said, when we cross-validated the $r^2$ performance on the higher-order moments, the results were not significantly higher than 0 for the CB mixture, and as such, accurately estimating higher order moments requires larger datasets than what we have considered here.

Nevertheless, in spite of the inability to capture predictive performance on these moments, we speculate that when combined across higher order moments and cross-moments, the ability of mixture models to capture higher order structure in the data is necessary for maximizing decoding performance, and that these moments might play an important role in neural coding. As the complexity of neural datasets increases, a careful study of such higher-order statistics would become both feasible and warranted, and our mixture model approach could prove to be a useful tool in such work.

## Appendix 3

### Sample complexity of conditional mixtures

To develop a sense of the sample complexity of CMs, we repeated the cross-validation simulations with discrete CMs on subsets of our two datasets (see Conditional mixtures effectively model neural responses in macaque V1 and Conditional mixtures facilitate accurate and efficient decoding of neural responses). In particular, we ran a 10-fold cross-validation simulation on a single subsample of 25%, 50%, 75%, and 100%, of each of our datasets. On our anaesthetized dataset this occasionally resulted in some neurons recording 0 spikes in a giving training set, which tends to cause our training algorithm to diverge, and so we filtered out neurons with low firing rates, leaving 50 neurons in our anaesthetized dataset.

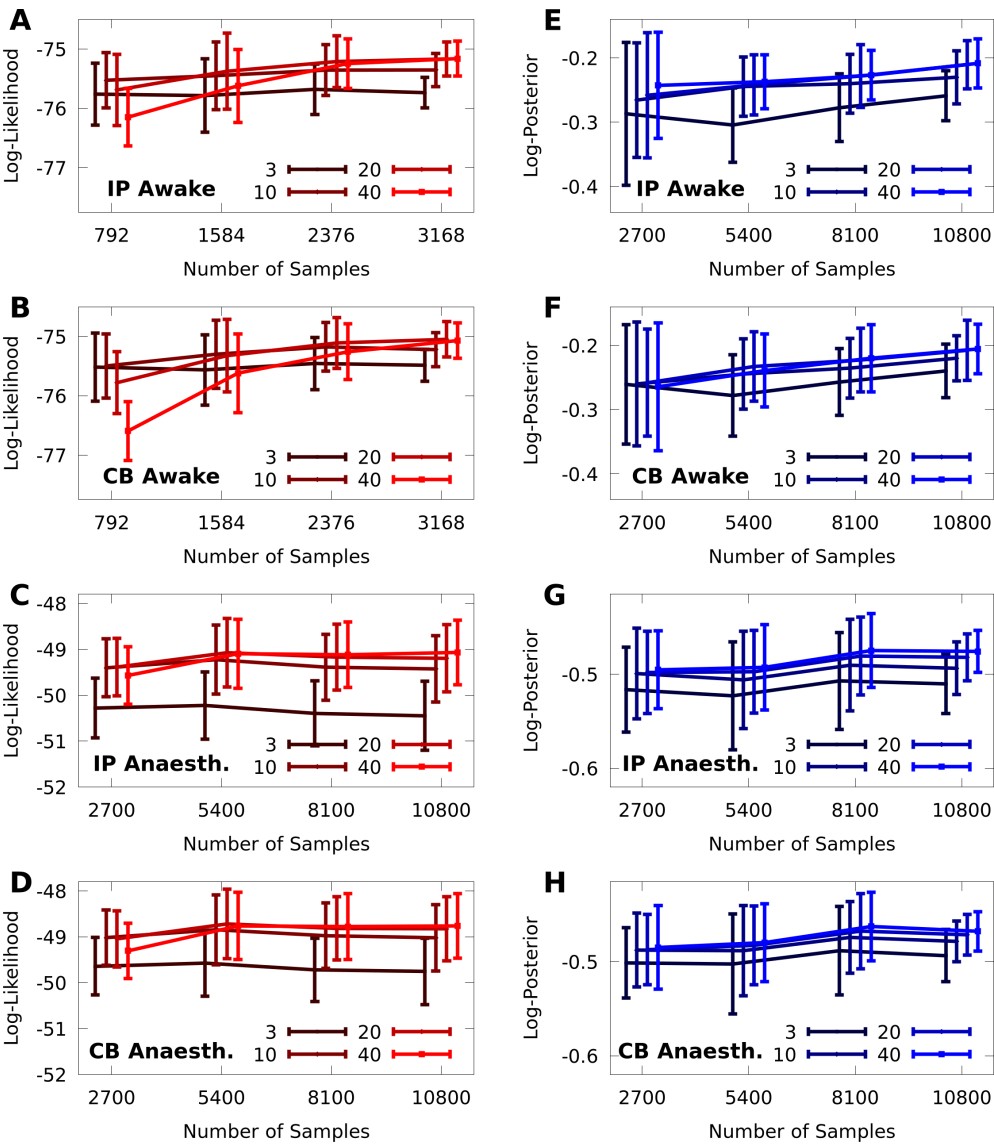

**Appendix 3—figure 1.** Sample complexity of discrete CMs. 10-fold cross-validation of discrete CMs with 3, 10, 20, and 40 components, on subsamples of 25%, 50%, 75%, and 100% of the awake and anaesthetized datasets, with $d_N = 43$ and $d_N = 50$ neurons, respectively. *Left column*: Cross-validated average log-likelihood of the models given test data (i.e. encoding performance). *Right column*: Cross-validated average log-posterior (i.e. decoding performance). Error bars represent standard error. In all panels, we added an offset on the abscissa for better visualization of the error bars.

We present the results of our simulations in *Figure 1*. In the left column (*Figure 1A–D*), we present the cross-validated log-likelihood of the vanilla and CoM-models, on the awake and anaesthetized data, respectively, and we see that, as we would expect, models with fewer components maximize their performance on smaller datasets. Even with large amounts of data, however, the performance difference between models with more than 10 components is nearly statistically indistinguishable. In the right column (*Figure 1E–H*), we present the cross-validated log-posterior performance of the models, and the results mirror those of the log-likelihood simulations, except the benefits of larger models becomes even more marginal.

## Appendix 4

### Conditional mixtures and generalized linear models

CMs are closely related to generalized linear models (GLMs), which are widely applied in neuroscience. The application of GLMs to modelling stimulus-driven spike-trains was pioneered in *Pillow et al., 2008*, in which the authors develop a Poisson encoding model $p(n_t \mid \mathbf{x}, \mathbf{m}_1, \ldots, \mathbf{m}_{d_N}) \propto e^{\theta_N(\mathbf{x}, \mathbf{m}_1, \ldots, \mathbf{m}_{d_N}) n_t}$, where $d_N$ is the number of recorded neurons, $n_t$ is the spike-count of the modelled neuron in timebin $t$, $\mathbf{x}$ is the stimulus (here the stimulus is an image and represented as a vector), and where each $\mathbf{m}_i$ is the spike-count history of the $i$ th recorded neuron up to time $t - 1$. The log-rate $\theta_N$ of the modelled neuron at time $t$ depends linearly on the stimulus and the spike-history, and is given by

$$\theta_N(\mathbf{x}, \mathbf{m}_1, \ldots, \mathbf{m}_{d_N}) = \mathbf{x} \cdot \mathbf{k} + \sum_{i=1}^{d_N} \mathbf{h}_i \cdot \mathbf{m}_i, \tag{18}$$

where $\mathbf{k}$ and $\mathbf{h}_i$ are vectors; in *Pillow et al., 2008* both $\mathbf{k}$ and $\mathbf{h}_i$ are represented by basis functions with a manageable number of fittable parameters.

This model may be trivially combined with a CM in order to extend the GLM formulation with a latent source of shared-variability that affords analytic expressions for various quantities of interest. The definition of a CB-CM is $p(\mathbf{n}, k \mid x) \propto e^{\boldsymbol{\theta}_N(x) \cdot \mathbf{n} + \boldsymbol{\theta}_N^* \cdot \mathbf{lf}(\mathbf{n}) + \boldsymbol{\theta}_K \cdot \boldsymbol{\delta}(k) + \mathbf{n} \cdot \boldsymbol{\Theta}_{NK} \cdot \boldsymbol{\delta}(k)}$, and we may simply replace the variable $x$ with all the independent variables in the GLM formulation, namely $\mathbf{x}$ and $\mathbf{m}_1, \ldots, \mathbf{m}_{d_N}$, and define the baseline log-firing rates $\boldsymbol{\theta}_N(\mathbf{x}, \mathbf{m}_1, \ldots, \mathbf{m}_{d_N})$ as $d_N$ copies of the function defined by *Equation 18*, each with its own independent parameters. In principle, the expectation-maximization framework we have presented for training CMs can be directly applied to a model with this structure. That being said, choosing the right parameterization of $\mathbf{k}$ and $\mathbf{h}_1, \ldots, \mathbf{h}_{d_N}$ would pose a unique challenge in a combined GLM-CM model, and whether such a model would be practically useful is an empirical question that is beyond the scope of this paper.

Here, we also clarify that although a CM is closely related to both mixture models and GLMs, it should not be confused with the models known as 'mixtures of GLMs' (*Grün and Leisch, 2008*). A mixture of GLMs has the form $p(y \mid x) = \sum_{k=1}^{d_K} w_k p(y \mid x; \boldsymbol{\theta}_k)$, where $d_K$ is the number of GLMs to be mixed, $w_k$ are the weight parameters, and $p(y \mid x; \boldsymbol{\theta}_k)$ is a GLM with parameters $\boldsymbol{\theta}_k$. This model differs from a CM in many subtle ways, and the easiest to note is that the weights $w_k$ do not depend on the stimulus $x$ as they do in a CM, which, as shown in *Figures 4* and *6* of the main paper, is critical to how CMs represent data.

