## [Decision Letter]

**Acceptance summary:**

This paper introduces a new framework for modeling correlated neural population spike responses using multivariate mixtures of Poisson or Conway-Maxwell-Poisson distributions. It describes an algorithm for fitting the model to data using Expectation Maximization (EM), a formula for Fisher information, and a Bayesian decoder that is competitive with other more computationally demanding decoding methods such as artificial neural networks. The authors apply this model to V1 data from awake and anesthetized monkeys, and show that it captures the variability (eg., Fano Factor) and co-variability of population responses better than Poisson models. Finally, the paper shows how the latent variables of the model can provide insight into the structure of population codes. The resulting framework represents a powerful advance for modeling the correlated variability in neural population responses, and promises to be a useful new tool for analyzing large-scale neural recordings. The paper will be of interest to computational neuroscientists studying neural coding, and to system neuroscientists who use descriptive models to characterize the stimulus tuning of correlated spiking activity recorded from large neural populations.

**Decision letter after peer review:**

Thank you for submitting your article "Modelling the neural code in large populations of correlated neurons" for consideration by *eLife*. Your article has been reviewed by 3 peer reviewers, and the evaluation has been overseen by a Reviewing Editor and Joshua Gold as the Senior Editor. The following individual involved in review of your submission has agreed to reveal their identity: Kenneth D Harris (Reviewer #3).

1) Clarify abstract (and optionally title): see comments from R1.

2) Address R1 comments about claims "C1" and "C2".

3) Compare performance of the proposed models to existing models that capture stimulus encoding by large populations of correlated neurons (see comments from R2).

4) Clarify how users should decide how many mixture components to add. (R3 comment 4).

5) Provide publicly released code.

*Reviewer #1 (Recommendations for the authors):*

1 – A main suggestion is to rewrite the title and abstract as I outlined in Weakness #1 in the public comments.

2 – Address 2.1-2.3 (under weaknesses), in the public comments or eliminate/tone down the claims C1-C3, in the abstract, intro, discussion or elsewhere.

3 – I found the discussion around Figure 5 hard to follow. To improve this, I suggest that authors provide (already in the Results section) simplified/readable formulas (valid for the "vanilla" Poisson case which is used in this section) for the stimulus dependence of index probabilities (IP) which depends on the mean population response according to

p(k|x) ∝expθ_K_(k) + r(x|k)

where r(x|k) is the average total population spike count in the mixture component k, which I believe is given by the partition function ψ_N_.

4 – As they point out, in their minimal model, the tuning curves (mean response) of neurons are scaled ("gain modulated") versions of the "baseline"-component's tuning curves. Again, I think it would help to write a formula for this in the Results section, and connect the scale factor and the baseline tuning curve with the θ_NK_ and θ_N_(x) components, respectively.

5 – I don't think the simple scaling (see previous comment) relationship actually holds in the non-Vanilla CoM case, but they do claim that. If so, the text in lines 189-192 (especially "scaled" in line 191) should be corrected.

6 – As they derive in the Methods part, the population covariance matrix of the CPM has the same form as the covariance matrix in factor analysis: a diagonal "private noise" matrix + low-rank "shared-noise" matrix. I think it would be valuable to point this out and write the corresponding formula in the Results section e.g. around Figure 2. Also point out what happens to the diagonal term in the vanilla vs. CoM cases.

7 – The ground truth exercise of Figure 3 is valuable, but I think more valuable than showing how the model fits one example would be to give an idea of the "sample complexity": give an idea of goodness of fit vs. number of trials in the dataset. (At least clarify in the caption how many trials per stimulus conditions were used.)

8 – Not sure what the exercise described in lines 310-321 shows. Given that the gound truth model is within the fit model family, isn't it given (by classical asymptotic statistics results) that for large enough data the likelihood and therefore the posterior should converge the true posterior?

So is the result really surprising given that the dataset seems pretty large (d_T=10000)?

Again the more relevant thing would be: what is the minimal amount of data needed to find a good posterior approximation… or as a simpler version: how would it do for typical neural dataset sizes (# trials). (c.f. the previous comment).

9 – lines 359-371 – especially line 365-6: The reasoning here (that the shown results establish the information-limiting nature of the noise) are not really complete. Technically, "Information-limiting" means that the Fisher info is not extensive, i.e. does not scale linearly with population size. So they have to argue that the "random shifts" (discussed in line 365) will not go to zero as. Population size goes to infinity.

*Reviewer #3 (Recommendations for the authors):*

Line 103: Some more introduction to CoM Poisson distributions would be nice. Why are these better than the negative binomial, which is analytically more tractable? Presumably because they can handle underdispersion? Neural data is usually overdispersed, but does the extra dispersion introduced by a mixture model mean one needs to use underdispersed components for the mixture components?

Line 119: "express multivariate Poisson mixtures in an exponential family form". This is misleading: it sounds like you have expressed the marginal distribution of the mixture model in exponential form, which I believe is impossible. In fact, you are expressing each component distribution in exponential form.

Line 150: "vanilla mixtures". Why not call them Poisson mixtures? That's what they are.

Line 159: "optimized model parameters as described in Materials and methods". You mean you used the EM algorithm derived in Materials and methods? Say so explicitly.

Figure 2. Is this cross-validated? From the text it seems not, so no wonder the CoM model, with more parameters, fits better. Also, why does the vanilla model ever produce FFs that are too low? Can't it just add more mixture components to increase dispersion?

Line 180: the "CPM" sounds like a mixture of generalized linear models. If so, "mixture of GLMs" would be more familiar terminology for most readers.

Line 193-227: it is not clear what we really learn from this. If it just is a validation that the EM algorithm can work on simulated ground truth, then shouldn't that go first, before the application to real data? Also comparing to a less sophisticated model would help show the benefits of this one.

Table 1: please state how many cells are in both data sets.

Line 238: "log likelihood". Please specify if this is to base 2 or base e; also give a unit in table 1 (e.g. bits/trial).

Figure 5: it would be nice to see this applied to real data.

Line 466: do you mean ψN = ∑i θ_N,i_? The log partition functions should add, right?

Equation 12: is there a denominator of ∏_i_ n_i_! missing?

Line 573: how much time does the gradient ascent take? Is it going to be a problem for recordings with large numbers of neurons?

---

## [Author Response]

The reviewers have discussed their reviews with one another, and the Reviewing Editor has drafted this to help you prepare a revised submission.1) Clarify abstract (and optionally title): see comments from R1.

We rewrote the abstract to be more concrete, and state more clearly the nature of the model, and our exact contributions.

2) Address R1 comments about claims "C1" and "C2".

C1 concerned the exact relationship between our work and the theory of Probabilistic Population Codes. We clarified this relationship with a mathematical result and a new section in the manuscript (Constrained conditional mixtures support linear probabilistic population coding).

C2 concerned how we (over) described the “interpretability" of the latent variables in our application of conditional mixtures to synthetic data with information-limiting correlations. We reformulated relevant text to emphasize that the model in this case is descriptive and not mechanistic.

3) Compare performance of the proposed models to existing models that capture stimulus encoding by large populations of correlated neurons (see comments from R2).

We've expanded the article with an appendix, where we provided a thorough comparison of our model with factor analysis, and highlight these results where relevant in the main article. We focused our comparison on factor analysis since it has been applied to modelling unbounded, trial-to-trial spike-counts, and in our introduction we attempted to better explain this is the form of data that our model should be applied to. In the Introduction we have also included references to other models indicated by R2 (lines 44{51), and clarified how they differ from ours.

4) Clarify how users should decide how many mixture components to add. (R3 comment 4).

We added a subsection in the methods section (Strategies for choosing the CM form and latent structure) which presents strategies and suggestions for choosing the latent variable structure/number of mixture components when applying conditional mixtures.

5) Provide publicly released code.

Our code is now available at the Git repository: https://gitlab.com/sacha-sokoloski/

neural-mixtures, and at that page we have provided an overview of the library and instructions for running the code (we point to this page at lines 796-800). The code provides several scripts with which we generated the various figures in the manuscript, and we would be willing to write e.g. Python bindings if interested parties would desire more fine-grained control of the code and libraries.

Reviewer #1 (Recommendations for the authors):1 – A main suggestion is to rewrite the title and abstract as I outlined in Weakness #1 in the public comments.

See online response to public review.

2 – Address 2.1-2.3 (under weaknesses), in the public comments or eliminate/tone down the claims C1-C3, in the abstract, intro, discussion or elsewhere.

See online response to public review.

3 – I found the discussion around Figure 5 hard to follow. To improve this, I suggest that authors provide (already in the Results section) simplified/readable formulas (valid for the "vanilla" Poisson case which is used in this section) for the stimulus dependence of index probabilities (IP) which depends on the mean population response according to

p(k|x) ∝expθ_K_(k) + r(x|k)

where r(x|k) is the average total population spike count in the mixture component k, which I believe is given by the partition function ψ_N_.

We've added this formula (lines 415-416) and reworked this section (see the paragraphs from lines 413-434).

4 – As they point out, in their minimal model, the tuning curves (mean response) of neurons are scaled ("gain modulated") versions of the "baseline"-component's tuning curves. Again, I think it would help to write a formula for this in the Results section, and connect the scale factor and the baseline tuning curve with the θ_NK_ and θ_N_(x)components, respectively.

We now provide more explicit formulae that connect gain modulation with the exponential family parameters (lines 221-227).

5 – I don't think the simple scaling (see previous comment) relationship actually holds in the non-Vanilla CoM case, but they do claim that. If so, the text in lines 189-192 (especially "scaled" in line 191) should be corrected.

You are correct, and we've clarified the language in the corresponding section (lines 227-229).

6 – As they derive in the Methods part, the population covariance matrix of the CPM has the same form as the covariance matrix in factor analysis: a diagonal "private noise" matrix + low-rank "shared-noise" matrix. I think it would be valuable to point this out and write the corresponding formula in the Results section e.g. around Figure 2. Also point out what happens to the diagonal term in the vanilla vs. CoM cases.

This is very insightful, thank you. We've incorporated this comparison into the development and introduction of mixture our models (see lines 126-137, and 170-172). We've resisted adding the explicit formulas for the variances/covariances of the various models, as we found that it required introducing too much notation, however we hope we conveyed your insight effectively in words.

7 – The ground truth exercise of Figure 3 is valuable, but I think more valuable than showing how the model fits one example would be to give an idea of the "sample complexity": give an idea of goodness of fit vs. number of trials in the dataset. (At least clarify in the caption how many trials per stimulus conditions were used.)

We agree that this is important to establish. In Appendix 3 we repeated our cross-validated simulations for estimating the predictive log-likelihood and log-posterior on subsets of our two datasets, i.e. by subsampling different fractions of trials, to better understand the sample complexity of these two quantities.

8 – Not sure what the exercise described in lines 310-321 shows. Given that the gound truth model is within the fit model family, isn't it given (by classical asymptotic statistics results) that for large enough data the likelihood and therefore the posterior should converge the true posterior?So is the result really surprising given that the dataset seems pretty large (d_T=10000)?Again the more relevant thing would be: what is the minimal amount of data needed to find a good posterior approximation… or as a simpler version: how would it do for typical neural dataset sizes (# trials). (c.f. the previous comment).

Again we agree. We have removed this short paragraph and instead provide a sample complexity analysis in Appendix 3.

9 – lines 359-371 – especially line 365-6: The reasoning here (that the shown results establish the information-limiting nature of the noise) are not really complete. Technically, "Information-limiting" means that the Fisher info is not extensive, i.e. does not scale linearly with population size. So they have to argue that the "random shifts" (discussed in line 365) will not go to zero as. Population size goes to infinity.

This is a fair point, and we've reworded parts of this section to emphasize that our model learns to approximate information-limiting correlations, which we hope clarifies the difference. (See again the section starting at line 393, and in particular lines 392-393, and 435-436).

Reviewer #3 (Recommendations for the authors):Line 103: Some more introduction to CoM Poisson distributions would be nice. Why are these better than the negative binomial, which is analytically more tractable? Presumably because they can handle underdispersion? Neural data is usually overdispersed, but does the extra dispersion introduced by a mixture model mean one needs to use underdispersed components for the mixture components?

To address these questions, first, we added a bit more introduction to the CoM-Poisson distributions (lines 108-114), and indeed the reason we use them is to allow our components do exhibit under-dispersion, as described in Stevenson et al., 2016, Journal of Computational Neuroscience.

Secondly, as you say, it is indeed helpful to to use CoM-Poisson distributions even when the data is primarily over-dispersed, because mixture models can make the statistics “too" over-dispersed. A specific example of this effect is illustrated in our applications to anaesthetized data (Figure 2B). We explained this more concretely when we show the improvements that result from our CoM-based model, and show how the CoM-Based parameters facilitate this in Figures 2C-D.

Line 119: "express multivariate Poisson mixtures in an exponential family form". This is misleading: it sounds like you have expressed the marginal distribution of the mixture model in exponential form, which I believe is impossible. In fact, you are expressing each component distribution in exponential form.

Indeed, thanks for pointing this out. We rewrote the text to properly explain that mixtures are marginals of an exponential family joint distribution (lines 138{139).

Line 150: "vanilla mixtures". Why not call them Poisson mixtures? That's what they are.

Indeed, we realize now that our naming conventions could lead to confusion, and we have retooled the names and acronyms in the paper. We now refer to Independent Poisson (IP) and CoM-Based (CB) models, and for Conditional Mixtures we write CM, so we have IP and CB mixtures, and then IP-CMs and CB-CMs.

Line 159: "optimized model parameters as described in Materials and methods". You mean you used the EM algorithm derived in Materials and methods? Say so explicitly.

Changed (lines 178-179).

Figure 2. Is this cross-validated? From the text it seems not, so no wonder the CoM model, with more parameters, fits better. Also, why does the vanilla model ever produce FFs that are too low? Can't it just add more mixture components to increase dispersion?

The analyses around Figure 2 are not cross-validated. We highlight that better in the revised text (lines 178-179).

The reviewer is correct that it's not a surprise that the CoM model, with more parameters, fits better. Our goal with this exercise instead is to illustrate how it fits better. Indeed, intuitively, one would expect that the vanilla mixture should capture anaesthetized data well, since it is primarily overdispersed, yet we find that this is not necessarily the case.

In the revised manuscript we have emphasized and explained this counter-intuitive result with the histograms of the CoM parameters around Figure 2 (which has now been split into two figures) and related text. In short, the means and variances of a Poisson mixture are coupled through shared parameters, and the CoM-based parameters break this coupling, and in particular allow the model to capture over-dispersion directly without comprising its estimate of the mean. This correction highlights the strength and importance of incorporating CoM-Poisson distributions into our models.

Line 180: the "CPM" sounds like a mixture of generalized linear models. If so, "mixture of GLMs" would be more familiar terminology for most readers.

This is a logical conclusion but in fact mixtures of GLMs are different from our models. In Appendix 4 material we include an explanation of the relationship between these two models.

Line 193-227: it is not clear what we really learn from this. If it just is a validation that the EM algorithm can work on simulated ground truth, then shouldn't that go first, before the application to real data? Also comparing to a less sophisticated model would help show the benefits of this one.

This is a fair point. In some sense this is just a validation of the EM algorithm, however, in the first set of results around figures 2-3, before Line 193-227 (now Lines 230-263), we aren't considering stimuli, and the EM algorithm for stimulus-dependence, requires a further extension.

The paper is centred around two results - incorporating CoM-based parameters, and adding stimulus-dependence - and in some sense it's arbitrary which we start with. We went with the former, and we hope it's okay to leave the structure as is.

As for the comparison with less sophisticated models, in our cross-validation of the log-likelihood and log-posterior under various models, we do consider an independent Poisson model and a Linear decoder. In the text we have attempted to better explain that this is a “sanity check" and not a rigorous performance analysis of the model, which comes in later sections (lines 263-264).

Table 1: please state how many cells are in both data sets.

Done (Now Table 2).

Line 238: "log likelihood". Please specify if this is to base 2 or base e; also give a unit in table 1 (e.g. bits/trial).

Added and added (see Table 2, and line 276).

Figure 5: it would be nice to see this applied to real data.

We agree, but, as for the application on other datasets, we believe that a proper analysis of the latent structure of information-limiting correlations in real data deserves a dedicated and extensive study.

Line 466: do you mean ψN = ∑i θ_N,i_? The log partition functions should add, right?

Fixed, thank you (Line 542).

Equation 12: is there a denominator of ∏_i_ n_i_! missing?

Fixed, thank you (Now Equation 13).

Line 573: how much time does the gradient ascent take? Is it going to be a problem for recordings with large numbers of neurons?

The model scales quite well. The largest computation in the descent step is an outer product computation which is O(k * n), where k is the number of mixture components, n is the number of neurons. Nevertheless, there are subtleties (in particular around minibatch sizes and number of stimuli). We believe the best place for this information is in our gitlab repository, and so we have included it there.

For reference, we have successfully applied the model in synthetic experiments to populations of thousands of neurons, with training times on the order of hours. For tens of thousands there might be a need for refinement of our code, perhaps including GPU computation.